# Addressing Signal Delay in Deep Reinforcement Learning

**Wei Wang**[1] **Dongqi Han**[2] **Xufang Luo**[2] **Dongsheng Li**[2]
[1]Western University, Canada   [2]Microsoft Research Asia
waybaba2ww@gmail.com,{dongqihan, xufluo, dongsli}@microsoft.com

## Abstract

Despite the notable advancements in deep reinforcement learning (DRL) in recent years, a prevalent issue that is often overlooked is the impact of signal delay. Signal delay occurs when there is a lag between an agent's perception of the environment and its corresponding actions. In this paper, we first formalize delayed-observation Markov decision processes (DOMDP) by extending the standard MDP framework to incorporate signal delays. Next, we elucidate the challenges posed by the presence of signal delay in DRL, showing that trivial DRL algorithms and generic methods for partially observable tasks suffer greatly from delays. Lastly, we propose effective strategies to overcome these challenges. Our methods achieve remarkable performance in continuous robotic control tasks with large delays, yielding results comparable to those in non-delayed cases. Overall, our work contributes to a deeper understanding of DRL in the presence of signal delays and introduces novel approaches to address the associated challenges.

## 1 Introduction

Deep reinforcement learning (DRL) and its applications have undergone rapid development in recent years (Sutton & Barto, 1998). The success of DRL has been witnessed not only in virtual tasks like videos games (Vinyals et al., 2019) and simulated robotic environments (Haarnoja et al., 2018a), but also in many challenging real-world tasks such as controlling tokamaks (Degrave et al., 2022) and tuning language models with human feedback (Schulman et al., 2017; Brown et al., 2020).

However, an often existed problem has long been ignored in deep RL studies, that is the delay of signals, i.e., the agent may not immediately observe current environmental state, or the agent's action cannot immediately act on the environment. Signal delay exists widely in various practical applications. For example, in autonomous vehicle navigation (Jafaripournimchahi et al., 2022), delayed feedback can occur due to real-world constraints, such as network latency or sensor processing time. In finance (Fang et al., 2021), high-frequency trading algorithms may experience delays in receiving information about market conditions due to network congestion or data processing bottlenecks. In robotics, the communication latency between a robot's sensors, actuators, and control systems can lead to delayed responses (Abadía et al., 2021). Additionally, in medical applications, such as telemedicine or remote surgery (Meng et al., 2004), there can be a delay between receiving patient data and the actual execution of the required actions. Last but not least, sometimes even the delay is short (e.g., 1 ms due to neural network inference), the environment may have already changed a lot, e.g., when controlling a Tokamak (characteristic timescale of 0.1 ms) (Degrave et al., 2022). These delays pose a significant impact on the effectiveness of deep RL-based solutions, necessitating urgent research to address this challenge.

On the other hand, signal delay is also a critical issue in biological systems. For instance, neural signals in humans take approximately 150 ms (Gerwig et al., 2005) to propagate from the brain to the muscles (efferent delay) and from sensors to the brain (afferent delay). This delay can be significant in motor control (Bastian, 2006), considering that Usain Bolt moves for more than 10 meters and a pianist plays dozens of notes every second. As DRL is well recognized as a biologically plausible

---

This work was done during Wei Wang's internship at MSRA. Correspondence to: Dongqi Han (dongqihan@microsoft.com; Dongsheng Li (dongsli@microsoft.com).

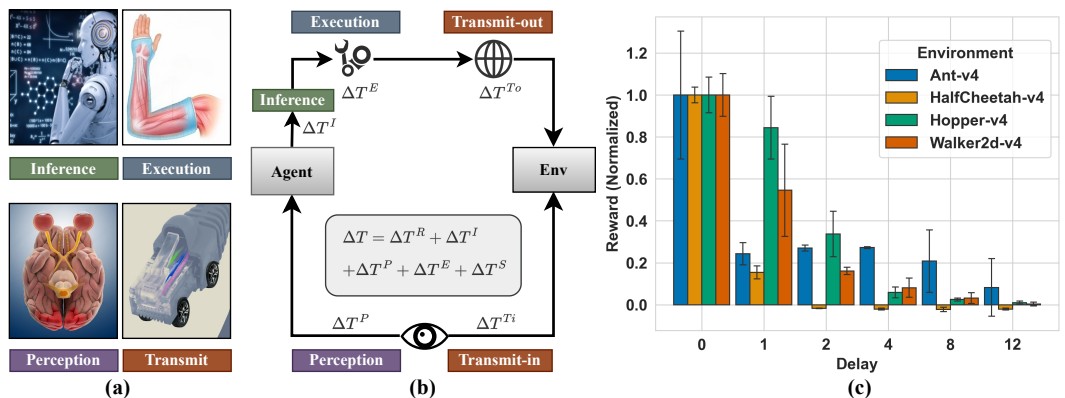

Figure 1: (a-b) Examples and illustration of delays in real-world scenarios. **Inference**: The process of inference incurs time costs (e.g., human brain, GPU). **Execution**: Time lags occur when signals travel to their point of execution (e.g., hand, robotic arm). **Perception**: Processing raw signals involves processing time (e.g., from eye to brain, sensor to processor). **Transmission**: Implementing actions often entails transmission delays to the intended environment (e.g., online gaming, remote control scenarios). (c) Even a few steps of delay significantly deteriorate the performance of SAC.

framework for modeling control and behavior (Botvinick et al., 2020), addressing signal delay in DRL could shed light on the corresponding neural mechanisms in animals and humans (Stein, 2009).

However, given the wide existence of signal delay and its importance, there is surprisingly few studies try to address this problem in DRL – while there have been extensive studies on delayed feedback in control theory and bandit problem, the only study, to our knowledge, explicitly touched this problem in deep reinforcement learning is from Chen et. al. (Chen et al., 2021). However, they make a strong assumption of knowing the reward function, which is often unknown in practice, and only discuss the case of fixed time of delay. The problem of signal delay in DRL in more general cases urges to be studied.

The current study aims to address the problem of deep RL with delay (DRLwD), particularly in continuous control tasks since most applications require continuous action space, such as robotics, autonomous driving, remote surgery, and modeling biological motor actions. The primary contributions of this work are summarized as follows. First, we investigate the impact of delay on performance and provide empirical evidence demonstrating its significant effect through comprehensive experiments. Next, we provide a mathematical formulation for the delay problem, encompassing both action and observation delays, and derive theoretical insights to devise an effective solution. Then, building on our suggested insights, we examine a range of ideas to mitigate or overcome the key challenges and provide an empirical evaluation of the effectiveness of each idea. Finally, we suggest simple and general approaches for actor-critic architectures that effectively address the impacts of signal delay in DRL. Overall, our work contributes to a deeper understanding of DRL in the presence of signal delays and presents a novel model architecture to overcome the associated challenges.

## 2 PROBLEM DEFINITION

### 2.1 BACKGROUND: PARTIALLY OBSERVABLE MDP

A POMDP is defined as a tuple $(\boldsymbol{S}, \boldsymbol{A}, \mathcal{P}_0, \mathcal{T}, \boldsymbol{X}, \mathcal{O}, \gamma)$, where $\boldsymbol{S}$ and $\boldsymbol{A}$ are the state and action spaces, respectively. $\mathcal{P}_0$ specifies the initial state distribution such that $\mathcal{P}_0(s)$ is the probability of a state $s \in \boldsymbol{S}$ being an initial state. $\mathcal{T}$ specifies the state transition probability such that $\mathcal{T}(s', r|s, a)$ is the probability of reaching to a new state $s' \in \boldsymbol{S}$ with an immediate reward $r \in \mathbb{R}$ after taking an action $a \in \mathcal{A}$ at a state $s \in \boldsymbol{S}$. $\boldsymbol{X}$ denotes the observation space. $\mathcal{O}$ specifies the observation probability such that $\mathcal{O}(x|s)$ is the probability of an observation $x \in \boldsymbol{X}$ at a state $s \in \boldsymbol{S}$. $\gamma \in [0, 1)$ is the discount factor.

## 2.2 DEFINING DELAYED-OBSERVATION MDP

First we consider a standard MDP without delay (referred to as the original MDP), in which state is denoted by $s$. We extend the original MDP to incorporate observation delay (delay steps can be non-fixed). We define a *Delayed observation Markov decision processes* (DOMDP)[1], as a special case of POMDP, by considering its state $\sigma = (s^{(-T)}, s^{(-T+1)}, ..., s^{(-1)}, s)$, where $T$ is the maximum delay, and $s$ is the state of the original MDP. Intuitively, $s^{(-t)}$ is the state of the original MDP from $t$ steps ago, and the superscript $(-t)$ indicates the relative timestep shift so as to maintain the Markovian property. The transition probability function of a DOMDP is then defined as $\mathcal{T}(\sigma', r|\sigma, a) = \mathcal{T}_0(s', r|s, a) \prod_{t=1}^{T} \mathbb{I}(s'^{(-t)} = s^{(-t+1)})$, where $\mathbb{I}$ denotes the indicator function, and $\mathcal{T}_0(s', r|s, a)$ is the transition probability of the original MDP. The term $\prod_{t=1}^{T} \mathbb{I}(s'^{(-t)} = s^{(-t+1)})$ transfers $s^{(-t+1)}$ at current step to $s^{(-t)}$ at next step, therefore explaining $s^{(-t)}$ as the delayed state $t$ from steps ago, without the necessity of introducing absolute time step as in a time-dependent MDP (Boyan & Littman, 2000).

Then we define the observation probability as $\mathcal{O}(\tilde{s}|\sigma) = \mathcal{O}(\tilde{s}|s^{(-T)}, s^{(-T+1)}, ..., s^{(-1)}, s) = \sum_{t=1}^{T} \mathcal{P}(t)\mathbb{I}(\tilde{s} = s^{(-t)})$, where $\tilde{s}$ is the observation (delayed state), and $P(t)$ is the probability that the signal delays for $t$ steps. A simple case is that $\mathcal{P}(t) = \mathbb{I}(t = \Delta T)$, which means the signal delay is fixed as $\Delta T$ steps, and thus $\tilde{s} = s^{(-\Delta T)}$. So far, we have defined a DOMDP by specifying elements of a POMDP.

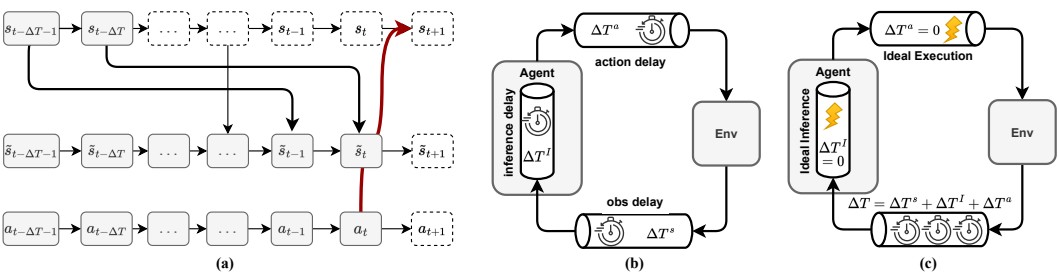

(a)  (b)  (c)

Figure 2: (a) Illustration of action and observation shifting in DOMDP. All kinds of delays (b) Delays examples, Inference Delay: the model inference time cost, Observation Delay: time cost to transition observation to the agent. Action delay: time to excute action. (c) Equivalent to the single delays with the same total durat.

The complexity and diversity of environments can lead to various kinds of delays, such as inference delay $\Delta T^a$, action delay $\Delta T^I$, and observation delay $\Delta T^s$, as showed in Fig. 2(b). largely adding to the intricacy of the problem. Thankfully, all these types of delays are equivalent. Only the sum of them, denoted by $\Delta T$, impacts the decision-making. This realization paves the way for a streamlined approach to modeling the problem. We proceed to establish this through the following theorem (Katsikopoulos & Engelbrecht, 2003):

**Theorem 2.1** (Delay Equivalence). *For an agent employing policy $\pi(a|\cdot)$ at time step $t$, the impact of state delay $\Delta T^s$, $\Delta T^I$ and $\Delta T^a$ on the observation transition $\tilde{\mathcal{P}}(\tilde{s}_{t+1}|\tilde{s}_t)$ is equivalent. In other words, as long as $\Delta T$ is the same, the agent perceives different variants of $\Delta T^s$, $\Delta T_I$ and $\Delta T_a$ values as identical. This conclusion hold for both the fixed and unfixed delay. (Proof in Appendix).*

Consequently, we focus on the DOMDP in right side of Fig. 2b for the sake of notational simplicity and comprehension. Specifically, in Fig. 2, the system is modeled by an ideal agent with zero inference time and zero execution time. The only delay between the state of the environment and the decision-making input of the agent is $\Delta T = \Delta T^s + \Delta T^a + \Delta T^I$. Therefore, we consider observation delay in this work without loss of generality.

---

[1]Note that Chen et al. (2021) has defined an delay-aware MDP. However, they assume the delay is fixed. By contrast, our definition considers a more general case where the delay could be variant and probabilistic.

## 3 CHALLENGE OF DRL WITH DELAYED SIGNALS

**Standard DRL and general POMDP algorithms exhibit catastrophic failure with signal delay**
Now we empirically examine the performance of existing RL algorithms straightforwardly applying
to environments subjected to various delays on four MuJoCo environments. The algorithms include
deep deterministic policy gradient (DDPG) (Lillicrap et al., 2015), twin-delayed DDPG (TD3) (Fu-
jimoto et al., 2018), soft actor-critic (Haarnoja et al., 2018b) and a RNN-based approach for general
POMDP tasks (RNN Strong) (Ni et al., 2022). Fig. 3 shows the normalized results of the exper-
iments, in which the reported scores are proportionate to the maximum attainable score for each
environment of SAC without delay. It reveals that even a modest delay of one environment step
substantially undermines the performance of the all these algorithms. Delays surpassing 4 steps lead
to catastrophic failure. These results underscore the considerable adverse impact delays can have
on the efficacy of DRL algorithms when being directly applied. In the follows, we try to figure out
where the difficulties of DRLwD stem from.

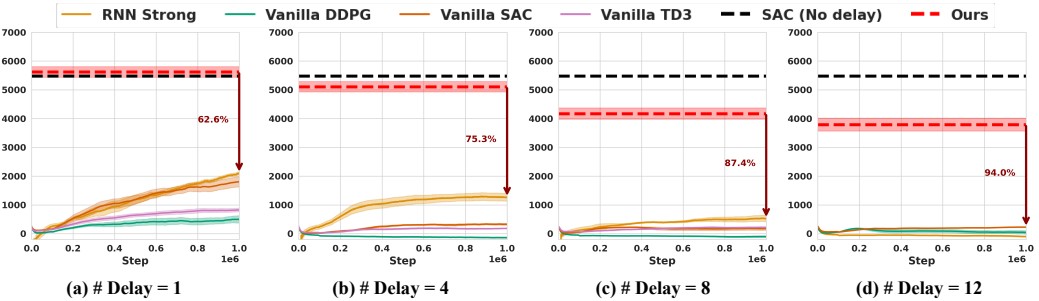

Figure 3: Task-averaged performance in MuJoCo environments (Ant-v4, HalfCheetah-v4, Hopper-
v4, Walker2d-v4) under various delay steps for state-of-the-art algorithms. SAC (No delay) acts as
the benchmark for optimal performance. The red dashed line and shaded area represent the mean
and standard deviation of our method's best-performing variants, serving as a benchmark.

**Misalignment between observed and true state** Fig. 4
illustrates that when observation signal delays occur, the
agent's actual state can differ from its observed state,
leading to varying optimal actions. Addressing this mis-
alignment between the current observation (delayed sig-
nal) and true state (non-delayed) is a central challenge in
DRLwD for optimal decisions. One straightforward solu-
tion is to estimate the true state based on available infor-
mation, such as the delayed signal and previous actions.
This would transform the problem to a standard MDP task
using the estimated true state. However, accurate estima-
tion can be difficult due to several factors, including com-
plex state transitions, stochastic environmental state tran-
sitions, and data distribution shifts (Pan & Yang, 2010)
during the process of online RL as the agent improves its

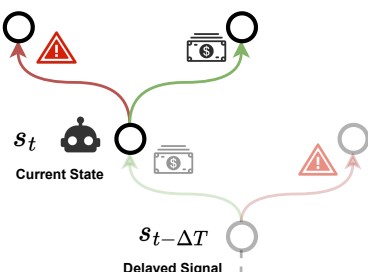

Figure 4: Varying optimal actions be-
tween actual state and observed state.

policy (Quinonero-Candela et al., 2008). In light of these challenges, alternative approaches are
needed to address the problem. The following section will discuss various methods, including our
proposed model, and empirically evaluate their effectiveness in handling the delay issue.

## 4 DEVELOPING ALGORITHMIC FRAMEWORK FOR DOMDP

Building on the definition and discussion presented earlier, our goal is to develop an algorithm that
effectively addresses the DOMDP problem. However, our objective extends beyond merely creating
a standalone algorithm specifically for DOMDP. Instead, we aim to explore a versatile framework
and components that can be integrated with current or future algorithms to handle DOMDP problems
effectively. This approach will lay a foundation that facilitates the seamless integration of future

developments in reinforcement learning with existing strategies. Considering the current state-of-the-art actor-critic methods has demonstrated success in continuous control tasks (Lillicrap et al., 2015; Haarnoja et al., 2018a; Fujimoto et al., 2018), we ground our work on actor-critic architecture.

The rest of this section is organized as follows: We first investigate the input design of the critic (Sec. 4.1) and the actor (Sec. 4.2). Subsequently, we explore other potential training techniques that show promise in improving performance under DOMDP scenarios (Sec. 4.3).

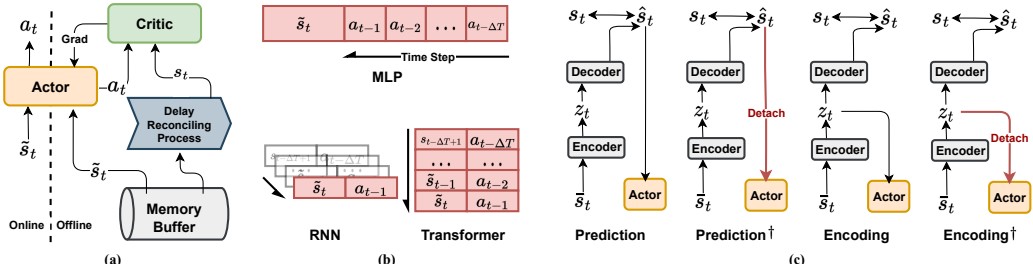

Figure 5: **Diagrams of the Proposed Methods:** (a) *Delay-Reconciled Training for Critic:* Utilizes only the online side during the deployment stage. The critic would use non-delayed information after recoiling in offline stage. (b) *State Augmentation for Actor:* Considers three methods for input $\bar{s}_t$ with historical information - MLP, RNN, and Transformer. For MLP, historical actions $a_{t-\Delta T:t-1}$ are concatenated with the state dimension. In RNN, the input consists of the current state and last-step action. In the case of the Transformer, the input includes state and action $\{s_i, a_{i-1}\}_{t-\Delta T:t-1}$ information spanning $\Delta T$ steps. (c) *Complementary Techniques for DOMDP Resolution:* Employs prediction/encoding auxiliary loss to aid in representation learning. An encoder first encodes the $\bar{s}_t$ into the hidden feature $z_t$, and then a decoder decodes it to $\hat{s}_t$. Prediction refers to using the predicted results $\hat{s}_t$ as the input of the actor, while the Encoding method utilizes features $z_t$ in the hidden space. For both methods, two variants are considered; *Detach* indicates stop of gradients backpropagation (with the notation of †).

## 4.1 DELAY-RECONCILED TRAINING FOR CRITIC

Estimating value functions is the most basic requirement for actor-critic methods. With time-shifted observations, a question naturally arises: how can we accurately estimate the state value, i.e., train the critic, under such circumstances? Contrary to standard POMDP problems, a crucial insight for DOMDPs is that full observation and reward is available during offline training once the delay is reconciled following real-time inference. In addition, as depicted in Fig. 5a, the forward pass of the critic is not required during the inference stage when interacting with the environment.

These facts motivate us to design a post-processing recovery pipeline that first recovers the historical information without delay, and then uses it for critic training. For instance, in delay-affected scenarios such as online gaming or trading, we can time-calibrate the historical data to recover the trajectory without delay. While this information is not available during inference, it can still be employed for critic training, thanks to the actor-critic structure. Similar techniques can be found in

Although previous studies, such as those by Vinyals et al. (2017); Baisero et al. (2022); Foerster et al. (2018), have demonstrated the efficacy of an asymmetric observation actor-critic paradigm. Research also indicates that an asymmetric input design may lead to a non-synchronous problem and gradient bias, as discussed by Baisero & Amato (2021). To further explore this issue, we include a comparison with a symmetric design in our study. In this alternative approach, the critic receives the same input as the actor, named with "Symmetric" as the prefix in the following context.

## 4.2 STATE AUGMENTATION FOR ACTOR

In the preceding section, we introduced a historical recovery pipeline for critic training. A similar approach, however, is not applicable to the actor, as the oracle $s_t$ is unavailable during the inference stage. This section explores potential ways to augment the actor's input to facilitate training.

Reviewing established POMDP algorithms, a common strategy, given the actor's inability to access the oracle state $s_t$ during inference, is to provide the actor with supplementary input to aid in the recovery of $\tilde{s}_{<t}$. These inputs typically consist of historical environmental observations such as opponents' behavior. However, it's evident that the historical state $\tilde{s}_{<t}$ is not informative in DOMDPs, as $I(s_t, \tilde{s}_{<t}|\tilde{s}_t) = 0$, assuming a Markovian environment, where $I(\cdot, \cdot)$ denotes mutual information.

So, aside from the historical state, what other information could aid in recovering the oracle state $s_t$? Intuitively, in DOMDP, as the oracle state $s_t$ is beyond $\tilde{s}_t$, the most significant transition influence is likely the actions taken during $s_t$ and $\tilde{s}_t$, denoted $\tilde{a}_{t-\Delta T:t}$. The following theorem (Katsikopoulos & Engelbrecht, 2003) proves the necessity of incorporating $\tilde{a}_{t-\Delta T:t}$ into the agent's input.

**Theorem 4.1** (Markovian Property). *Integrating historical actions $a_{<t}$ into the observation transforms it into an MDP with state transition probability $\bar{\mathcal{P}}(\bar{s}_{t+1}|\bar{s}_t, a_t)$, where $\bar{s}_t = (\tilde{s}_t, a_{t-\Delta T:t-1})$.*

The proof is deferred to Appendix D.2. The theoretical analysis suggests that in DOMDPs, the oracle state $s_{t_0}$ is independent of $\tilde{s}_{t<t_0}$ given $\tilde{s}_{t_0}$, i.e., $s_{t_0} \perp\!\!\!\perp \tilde{s}_{t<t_0}|\tilde{s}_{t_0}$. Additionally, it discloses a strong association between the oracle state $s_t$ and historical action $a_{t-\Delta T:t}$. In conclusion, incorporating historical actions into the observation allows us to convert a DOMDP to a MDP, which is a fundamental requirement for the convergence of many methods.

Next, we investigate two strategies to incorporate historical information into the actor, as in Fig. 5b: 1) MLP Encoder: Concatenating historical action data $a_{t-\Delta T:t}$ with $\tilde{s}_{t_0}$; 2) RNN Encoder: Adding only the last action $a_{t-1}$, more efficient for handling large action spaces or vast histories.

## 4.3 INVESTIGATION OF COMPLEMENTARY TECHNIQUES FOR DOMDP SOLUTION

Previous research (Igl et al., 2018; Subramanian et al., 2022; Lambrechts et al., 2023) shows that auxiliary loss can improve RL agent performance. Expanding on this and the Delay-Reconciled Training for Critic, our work introduces techniques beyond input design to enhance our framework. We aim to enable the actor to "imagining" the true state $s_t$ from delayed observations $\tilde{s}_t$ and actions $a_{t-\Delta T:t}$. Currently, the actor's learning relies on the critic's value estimations. To improve this, we propose two strategies using oracle observations for additional supervision.

**Prediction** Refer to Fig. 5c left side, a prediction network is trained, using $\bar{s}_t$ as input and $\hat{s}_t$ as output. The loss between the predicted state $\hat{s}_t$ and the oracle state $s_t$ is minimized. The prediction results are then utilized as actor input. Two variants were tested: one detaches (indicated with †) the output of the observation prediction network before inputting it to the policy network $\pi(\cdot)$, and the other does not. Detaching the variables can stabilize the interaction between prediction and policy networks while maintaining the connection potentially enhances supervision learning.

**Encoding** As depicted in Fig. 5c Right, a prediction network is trained to generate hidden features $z_t$. These features, $\tilde{z}_t$, serve as the input to the policy network $\pi(\cdot)$. This approach may be particularly effective when the original observation is hard to accurately predict, or when information is sparse.

## 5 EXPERIMENTAL RESULTS

We performed our experimental evaluations across MuJoCo environments (Todorov et al., 2012) with signal delay. We compare to popular RL algorithms for continuous including DDPG (Lillicrap et al., 2015), TD3 (Fujimoto et al., 2018), SAC (Haarnoja et al., 2018b); and DRL algorithms for POMDP including RNN-based (Ni et al., 2022) and belief-state-based (Han et al., 2020) (VRM); as well as DATS (Chen et al., 2021), a model-based algorithm for DRLwD (Tab. 1). We examine four distinct environment settings: (1) Fixed Delay, (2) Unfixed Delay, (3) Probabilistic State Transition (4) Large State Space. (1-3) include Ant-v4, Walker2d-v4, Hopper-v4 and HalfCheetah-v4.

**Implementation Details** are provided in the appendix, covering the environment setup (Sec. E.1), implementation of baselines from other research (Sec. E.2), our network architecture (Sec. E.3), hyperparameter selection (Sec. E.4), implementation of delayed environments (Sec. E.5), our code framework (Sec. E.6), and additional details (Sec. E.7).

## 5.1 EMPIRICAL EVALUATION OF DESIGNS

**Delay-Reconciled Training for Critic** The learning curve associated with the traditional method exhibits a plateau (Fig. 3), indicating that the learning process is not effectively progressing. However, when we apply the new design for the critic, the learning curves for both fixed (Fig. 6c) and unfixed delay scenarios (Fig. 6e) display a more dynamic and promising trend. This indicates that our Delay-Reconciled Training design contributes to a more progressive learning process. Furthermore, as demonstrated in Fig. 6a,b, the performance of Delay-Reconciled Training for the Critic consistently surpasses that of the Vanilla SAC when delay is introduced, and this performance gap widens as the delay increases. This result underscores the robustness of our design for the critic.

**State Augmentation for Actor** The aforementioned results demonstrate the effectiveness of our critic input design. Building upon this, we further apply state augmentation for the actor, based on the Delay-Reconciled Training for Critic. As shown in Fig. 6c,e, the learning curve of the model trained solely with Delay-Reconciled Training for Critic plateaus when the delay exceeds 8 steps. However, after incorporating state augmentation for the actor, the learning curve becomes more positive, as depicted in Fig. 6d,f. Moreover, the performance of the state augmentation for the actor consistently exceeds that of the model trained solely with Delay-Reconciled Training for Critic (Fig. 6a,b). As shown in Tab. 1, the final average improvement is $1.2\% \rightarrow 50.8\% \rightarrow 75.9\%$ on fixed delay and $8.4\% \rightarrow 48.1\% \rightarrow 77.0\%$ on unfixed delay. These results suggest that both techniques contribute to consistent performance gain and can be combined for further enhancement.

**Complementary Techniques.** As evident from Tab. 1, the most effective performance for each delay predominantly arises from algorithms employing Prediction and Encoding techniques. Specifically, when the delay is fixed and the delay step exceeds 1, the top-performing algorithms consistently involve Encoding[†] and Prediction[†]. On average, Encoding[†] and Prediction[†] enhance the performance of State Augmentation for Actor from 75.9% to 84.5% and 83.6%, respectively. However, the efficacy of explicit supervision significantly diminishes when the delay is unfixed. The average performance of Encoding[†] (77.9%) approximates that of State Augmentation (77.0%). Notably, the performance of Prediction[†] (72.5%) even drops by 4.5% compared to State Augmentation. This observation underscores that the explicit introduction of Prediction doesn't always guarantee performance improvement. Whether the delay is fixed substantially influences the effectiveness of techniques introducing additional prediction supervision.

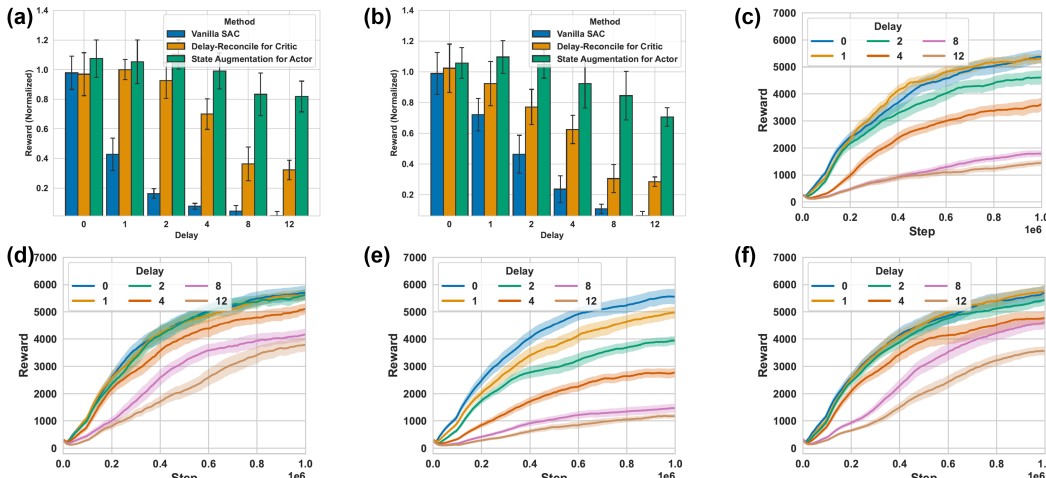

Figure 6: **Consistent performance improvement of critic and actor input design.** (a) Fixed Delay (b) Unfixed Delay (c) Delay-Reconciled Training for Critic with fixed delay (d) Delay-Reconciled Training for Critic with fixed delay (e) Actor State Augmentation- MLP with unfixed delay (f) Actor State Augmentation- MLP with unfixed delay. Rewards are averaged over four MuJoCo tasks.

Table 1: **Performance (%) of algorithms in fixed and unfixed delay environments.** The shaded values represent results from unfixed delay environments. All values are normalized based on the environment-specific performance of Vanilla SAC with no delay*. Data are presented as mean $\pm$ standard error of the mean (S.E.M). The best performing methods, including those within the range of S.E.M of the best, are highlighted in **bold**. Here, $s_t$ denotes the non-delayed state, $\tilde{s}_t = s_{t-\Delta T}$ represents the delayed state, and $\bar{s}_t = (\tilde{s}_t, a_{t-\Delta T:t-1})$ signifies the state with historical actions. $\hat{s}_t$ is the predicted state, while $z_t$ is the hidden feature of the prediction network. In. and Net. denote the input and network of the critic/actor, respectively.

| Method | Critic In. | Critic Net. | Actor In. | Actor Net. | 0 | 1 | | 2 | | 4 | | 8 | | 12 | | Avg. #≥4 | |
|---|---|---|---|---|---|---|---|---|---|---|---|---|---|---|---|---|---|
| | | | | | | | | | | # Delayed Time Steps | | | | | | | |
| *Existing Methods* | | | | | | | | | | | | | | | | | |
| Vanilla DDPG (Lillicrap et al., 2015) | $\tilde{s}_t$ | MLP | $\tilde{s}_t$ | MLP | 49.6 ±12.2 | 13.3 ±8.2 | 15.7 ±10.1 | 5.9 ±6.0 | 2.3 ±12.0 | -1.3 ±2.2 | 10.5 ±6.3 | -2.2 ±1.4 | -1.2 ±1.9 | -1.6 ±1.0 | -1.4 ±1.3 | -1.7 ±1.5 | 2.6 ±3.2 |
| Vanilla TD3 (Fujimoto et al., 2018) | $\tilde{s}_t$ | MLP | $\tilde{s}_t$ | MLP | **96.3 ±10.5** | 22.9 ±6.6 | 59.4 ±9.6 | 10.5 ±3.3 | 36.6 ±8.7 | 7.1 ±3.3 | 16.7 ±4.6 | 6.6 ±3.6 | 8.1 ±3.2 | 6.7 ±3.7 | 7.7 ±3.5 | 6.8 ±3.5 | 10.8 ±3.8 |
| Vanilla SAC (Haarnoja et al., 2018b) | $\tilde{s}_t$ | MLP | $\tilde{s}_t$ | MLP | **100.0* ±9.4** | 40.3 ±12.3 | 62.5 ±10.6 | 20.5 ±8.4 | 44.6 ±15.2 | 5.3 ±3.4 | 18.1 ±7.5 | -1.0 ±2.3 | 5.0 ±3.3 | -0.7 ±1.1 | 2.2 ±2.0 | 1.2 ±2.3 | 8.4 ±4.3 |
| RNN Strong Baseline (Ni et al., 2022) | $\tilde{s}_t$ | RNN | $\tilde{s}_t$ | RNN | 83.7 ±12.1 | 58.2 ±9.1 | 62.9 ±9.5 | 43.3 ±7.1 | 41.7 ±8.9 | 34.5 ±8.2 | 32.4 ±6.7 | 24.7 ±4.1 | 24.5 ±5.1 | 16.1 ±6.1 | 14.2 ±1.1 | 23.6 ±4.5 | 23.7 ±4.3 |
| VRM (Han et al., 2020) | $\tilde{s}_t$ | RNN | $\tilde{s}_t$ | RNN | 91.1 ± 16.7 | 64.8 ± 12.3 | 71.5 ±20.7 | 54.9 ±17.6 | 62.7 ±14.9 | 32.2 ± 12.0 | 41.4 ±16.7 | 21.4 ± 5.1 | 29.7 ±10.0 | 14.73 ± 2.2 | 16.4 ±2.0 | 22.7 ±6.4 | 29.1 ±9.5 |
| DATS (Chen et al., 2021) | $\tilde{s}_t$ | RNN | $\tilde{s}_t$ | RNN | 98.7 ± 13.4 | 73.9 ± 11.3 | 70.9 ±13.2 | 60.7 ±12.7 | 55.0 ±9.7 | 50.3 ±8.2 | 34.7 ±8.1 | 22.9 ± 4.7 | 28.4 ± 4.2 | 19.1 ± 5.1 | 14.3 ± 1.1 | 30.8 ± 6.0 | 25.8 ± 4.5 |
| *Critic Input Design* | | | | | | | | | | | | | | | | | |
| Delay-Reconciled Training | $s_t$ | MLP | $\tilde{s}_t$ | MLP | **99.2 ±8.0** | 92.9 ±10.0 | 94.9 ±8.5 | 84.5 ±10.4 | 85.0 ±10.8 | 73.5 ±11.1 | 65.7 ±11.7 | 46.7 ±9.7 | 46.9 ±10.5 | 32.3 ±9.1 | 31.8 ±8.8 | 50.8 ±9.9 | 48.1 ±10.3 |
| *Actor Input Design* | | | | | | | | | | | | | | | | | |
| State Augmentation - MLP | $s_t$ | MLP | $\bar{s}_t$ | MLP | **100.8 ±11.2** | 92.8 ±7.1 | 96.6 ±6.6 | 93.4 ±8.1 | **102.1 ±13.1** | 78.0 ±9.2 | 81.6 ±9.6 | 85.0 ±9.9 | 84.0 ±9.9 | 64.7 ±8.6 | 65.5 ±8.4 | 75.9 ±9.2 | 77.0 ±9.3 |
| State Augmentation - RNN | $s_t$ | MLP | $\bar{s}_t$ | RNN | 84.4 ±11.5 | 71.0 ±10.4 | 72.3 ±12.1 | 66.6 ±13.0 | 60.4 ±11.8 | 37.5 ±10.2 | 43.7 ±10.9 | 24.6 ±7.2 | 22.4 ±4.8 | 21.3 ±5.1 | 15.0 ±3.4 | 27.8 ±7.5 | 27.0 ±6.3 |
| State Augmentation - Transformer | $s_t$ | MLP | $\bar{s}_t$ | Tran. | 76.1 ±8.1 | 72.3 ±12.2 | 76.9 ±12.3 | 58.1 ±10.5 | 57.3 ±11.3 | 57.2 ±6.2 | 40.8 ±7.6 | 32.0 ±6.3 | 19.8 ±5.2 | 16.3 ±6.6 | 17.4 ± 5.7 | 35.1 ± 6.4 | 26.0 ± 6.2 |
| *Exploring Extended Techniques* | | | | | | | | | | | | | | | | | |
| Prediction† | $s_t$ | MLP | $\hat{s}_t$ | MLP | **102.1 ±9.7** | **101.5 ±13.9** | 96.1 ±12.2 | **100.5 ±14.4** | 99.0 ±12.5 | 92.2 ±16.2 | 88.6 ±14.4 | 85.6 ±10.8 | 71.0 ±12.8 | **73.0 ±10.3** | 58.1 ±14.1 | 83.6 ±12.5 | 72.5 ±13.8 |
| Encoding† | $s_t$ | MLP | $z_t$ | MLP | 97.6 ±12.1 | **101.4 ±9.0** | 113.0 ±9.6 | 103.2 ±10.0 | 92.0 ±9.5 | 90.4 ±9.5 | 86.6 ±8.2 | 89.9 ±12.4 | 77.9 ±10.8 | **73.3 ±9.9** | 69.2 ±9.7 | 84.5 ±10.6 | 77.9 ±9.6 |
| *Others* | | | | | | | | | | | | | | | | | |
| Symmetric - MLP | $\bar{s}_t$ | MLP | $\bar{s}_t$ | MLP | **106.3 ±9.4** | 75.1 ±9.1 | 72.5 ±8.9 | 63.7 ±9.7 | 61.2 ±9.0 | 51.4 ±11.7 | 52.0 ±9.6 | 37.5 ±13.2 | 31.0 ±7.0 | 27.0 ±12.0 | 28.3 ±5.0 | 38.6 ±12.3 | 37.1 ±7.2 |

## 5.2 PERFORMANCE ACROSS DIVERSE ENVIRONMENTS

**Performance on Basic Environments.** As shown in Tab. 1, DDPG, TD3, and SAC experience a significant performance drop of over 79.6% when the 2 exceeds four time steps. Even a single-step delay can cause DDPG's performance to plummet from 49.6% to 13.3% (fixed delay) and 15.7% (unfixed delay). TD3 and SAC exhibit a smaller yet substantial performance reduction, greater than 29.6%. While the RNN Strong (Ni et al., 2022) and VRM (Han et al., 2020) can somewhat function in POMDP and enhance SAC's performance, the improvement is relatively mild. Our simple Delay-Reconciled Training for Critic can considerably boost SAC's performance, consistently outperforming RNN Strong. The actor design also contributes to consistent performance improvement. For the techniques, we observe that the detached versions always outperform their non-detached counterparts; due to space constraints, we defer these results to Tab. 4 in the Appendix. For the symmetric design of the actor and critic which both use $\tilde{s}_t$ as input. The results reveal that this symmetric design (with an average performance of 37.1%) significantly underperforms compared to the asymmetric design with delay-reconcile critic. Additional investigations delve into the possible reasons of the baseline failures are provided, including issues related to representation learning (Sec. F.1) and redundant historical information (Sec. F.2). The learning curve can be found in Sec. F.7.

**Fixed v.s. Unfixed delay** As shown in Tab. 1, the stability of the delay most significantly influences prediction and encoding-based methods. Under fixed delay settings, prediction-based methods maintain 79% performance even with a 12-step delay, outperforming the second-best Actor State Augmentation-MLP at 61%. However, in an unfixed delay setting, the performance of prediction drops to 59%, falling below that of Actor State Augmentation-MLP (79%). This implies that explicitly introducing prediction supervision can have a detrimental effect on performance, influenced by whether the delay is fixed. This aligns with the intuition that future states become more unpredictable when delay steps vary. Additional results can be found in Sec. F.3 in the appendix.

**Performance under Large Observation Space and Probabilistic Environments.** The size of the observation space impacts the performance of algorithms that explicitly introduce supervised learning supervision. As depicted in Fig. 7a-d and Fig. 7f, the performance of all algorithms drops significantly as the number of delay steps increases. Meanwhile, the performance of the Actor State Augmentation- MLP is less severely affected (Fig. 7e), experiencing only around a 20% performance decrease when the number of delay steps reaches 12. Interestingly, when the prediction network is detached (Fig. 7b,d), the performance deteriorates even more drastically. This could possibly be attributed to the complexity of predicting true observations, leading the prediction network to generate less useful information for the policy network. Additional insights and analyses are provided in Sec. F.4 of the appendix. Probabilistic environments could also contribute to the under-performance of prediction methods, with more detailed findings available in Sec. F.5 and Sec. F.6.

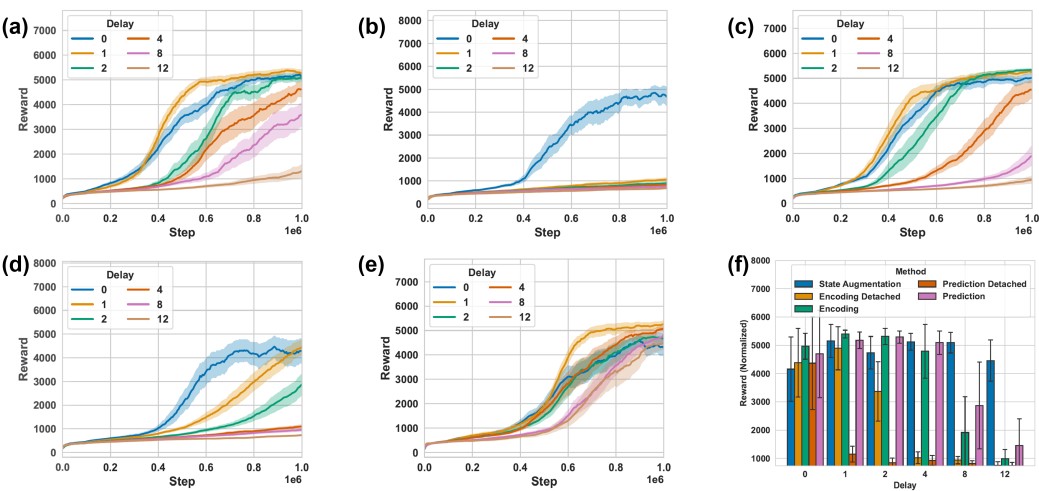

Figure 7: **Performance when observation space is large.** (a) Prediction (b) Prediction$^\dagger$ (c) Encoding (d) Encoding$^\dagger$ (e) Actor State Augmentation- MLP (f) Performance summary

### 5.3 TAKE-AWAY MESSAGE

We now summarize the main findings of our experiments: **(1)** The time recovery process presents a consistent and impactful enhancement to the performance of the Actor-Critic framework in DOMDP (Tab. 1). **(2)** Augmenting the observation to $\bar{s}_t$ can also consistently contribute positively towards overall performance (Tab. 1). **(3)** Introducing a prediction network (detached) can yield promising performance when the environment is simple (fixed part in Tab. 1). **(4)** Explicit prediction or model-based methods can lead to performance decrease (Tab. 1, Fig. 7) in scenarios where the environment is highly complex (large observation spaces - Fig. 7) and difficult to model (probabilistic - Fig. 7).

## 6 CONCLUSION

This work aims to address signal delay, a commonly overlooked issue in DRL (see Appendix B for a discussion about related works). We formulate DRLwD and propose effective approaches to alleviate the challenge of misalignment between observed and true states. Our research has certain limitations due to its focus on simulated robotic control environments. Dealing with signal delay in real-world applications could potentially encounter greater levels of uncertainty and additional challenges. Future work should explore real-world scenarios and transition from simulated to real environments (James et al., 2017; Kadian et al., 2020), for more discussions see Sec. C in Appendix.

ACKNOWLEDGEMENTS

This work was supported by Microsoft Research. The authors thank Yifei Shen for insightful contributions to the discussions that greatly enhanced this research.

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

## A  ETHICAL STATEMENT AND BROADER IMPACT

This work addresses signal delay in DRL with a commitment to transparency, reproducibility, and responsible application of our findings. The potential for misuse of our methods in harmful applications is acknowledged, and we advocate for their use within ethical and legal guidelines.

Our work has potential for substantial impact, enhancing efficiency and safety of autonomous systems reliant on DRL. However, the possibility of misuse in ethically ambiguous applications exists. Our standardization of DOMDP could guide future DRL research, but diverse focus within the field is necessary to avoid research biases. We encourage the use of this work responsibly, considering its ethical implications.

## B  RELATED WORK

**Signal delay in decision-making**   Several previous studies have dealt with signal delay in MDPs (Bander & White, 1999; Katsikopoulos & Engelbrecht, 2003; Walsh et al., 2009), however, their results have not been extended to deep RL. Chen et al. (2021) is the only work we found explicitly discuss DRLwD. There is also related work on traditional control and bandit problems with feedback delay. Recent works like Blanchet et al. (2023) and Wan et al. (2022) have tailored upper confidence bounds and Thompson sampling algorithms to handle unknown and stochastic delays in generalized linear contextual bandits. Li et al. (2019) explored the complexity of non-stochastic multi-armed bandit and bandit convex optimization settings with unknown delays. Verma et al. (2022) provided solutions for stochastic delayed feedback in Bayesian optimization, presenting novel algorithms with sub-linear regret guarantees for efficient new function query selection amidst random delays. A concurrent study (Kim et al., 2023) proposes an actor-critic algorithm which shows a great performance gain in DRLwD by compressing the dimension of the augmented state. Our work considers architecture design for DRLwD, which is orthogonal to these studies.

**POMDP methods**   Theoretical complexity analysis reveals that solving POMDPs can generally be both statistically and computationally challenging (Papadimitriou & Tsitsiklis, 1987; Vlassis et al., 2012). A common solution for POMDP is to model the function with additional context by employing RNNs, which are used to distill the characteristics of prior observations and actions into the states of the RNN (Schmidhuber, 1990; Jaderberg et al., 2019; Vinyals et al., 2019). While a DOMDP is an instance of the broader POMDPs, we have shown that by incorporating contextual actions into observation turns DOMDP to a MDP problem. Nonetheless, we examined the performance of an state-of-the-art RNN-based POMDP algorithm that performs good for many partially observable tasks (Ni et al., 2022), which is shown to be not suitable for DRLwD.

## C  FUTURE WORK AND DISCUSSIONS

In this section, we focus on the potential integration of timestamp data into our model, a concept that presents promising avenues for future research but also comes with its own set of challenges.

### C.1  ENHANCING MODEL ROBUSTNESS WITH TIMESTAMP DATA

The integration of timestamp data into our model is primarily aimed at optimizing performance in environments with variable delays and managing asynchronous data from multiple sources. This approach is particularly relevant when the historical number $H$ aligns with the delay $\Delta T$, as it can significantly improve the model's adaptability to changing delay patterns and enhance decision-making in systems with multiple, asynchronous data streams.

However, it's important to note that the effectiveness of this method may vary depending on the specific environment. For instance, certain sensors might not support timestamping, or network messages may lack time-related data. This makes the trick not a general solution for all cases

These challenges are the primary reason why we have earmarked the integration of timestamp data as a subject for future work rather than including it in the main body of our current paper. Future research will need to address these environmental variabilities and develop adaptable models that can effectively leverage timestamp data where available.

## C.2 Broadening the Application Beyond the Actor-Critic Framework

The issue of delayed signal can arise in a broad range of applications. Hence, it is crucial to develop methodologies that are sufficiently versatile to be integrated into various frameworks. In this project, we introduce a general approach that can be seamlessly incorporated into any actor-critic framework. While some algorithms do not adhere to the actor-critic framework. Expanding our methods to be entirely algorithm- and model-agnostic would significantly enhance their applicability and utility in a wider array of RL contexts.

## D Theorem Proof

### D.1 Theorem 2.1 Derivation

*Proof.* Let action delay be denoted as $\Delta T^a$ and observation delay as $\Delta T^s$. We denote the actual action acting on the environment at time step $t$ as $a_t$, and the action given by the agent at step $t$ as $\tilde{a}_t$. With action delay $\Delta T^a$, we have $a_t = \tilde{a}_{t-\Delta T^a}$. The inference delay $\Delta T^I$ can be considered included in either of the action and observation delays. From the agent's view, the state transition probability under policy $\pi(a|\cdot)$ with delay can be written as:

$$
\begin{aligned}
&\tilde{\mathcal{P}}(\tilde{s}_{t+1}|\tilde{s}_t) \\
&= \sum_{\Delta T^s=0}^{\Delta T^s_{\max}} \mathcal{P}(\Delta T^s)\mathcal{P}(s_{t+1-\Delta T^s}|s_{t-\Delta T^s}) \\
&= \sum_{\Delta T^s=0}^{\Delta T^s_{\max}} \int_{a_{t-\Delta T^s}} \mathcal{P}(\Delta T^s)\mathcal{P}(s_{t+1-\Delta T^s}|s_{t-\Delta T^s}, a_{t-\Delta T^s})\mathcal{P}(a_{t-\Delta T^s})da_{t-\Delta T^s} \\
&= \sum_{\Delta T^s=0}^{\Delta T^s_{\max}} \sum_{\Delta T^a=0}^{\Delta T^a_{\max}} \int_{\tilde{a}_{t-\Delta T^s-\Delta T^a}} \mathcal{P}(\Delta T^s, \Delta T^a)\mathcal{P}(s_{t+1-\Delta T^s}|s_{t-\Delta T^s}, \tilde{a}_{t-\Delta T^s-\Delta T^a}) \\
&\qquad\qquad\qquad\qquad\qquad\qquad \pi(\tilde{a}_{t-\Delta T^s-\Delta T^a}|I_{t-\Delta T^s-\Delta T^a})d\tilde{a}_{t-\Delta T^s-\Delta T^a}.
\end{aligned}
$$

Here, $\mathcal{P}(\Delta T^s)$ and $\mathcal{P}(\Delta T^a)$ denote the probability of state and action signals delay for certain steps. $I_{t-\Delta T^s-\Delta T^a}$ denotes the information available to the agent for selecting action $\tilde{a}_{t-\Delta T^s-\Delta T^a}$, which is until time step $t - \Delta T^s - \Delta T^a$.

Since the state transition probability function $\mathcal{P}(s_{t+1-\Delta T^s}|s_{t-\Delta T^s}, \tilde{a}_{t-\Delta T^s-\Delta T^a})$ is Markovian (time-independent), we have

$$
\begin{aligned}
&\tilde{\mathcal{P}}(\tilde{s}_{t+1}|\tilde{s}_t) \\
&= \sum_{\Delta T^s=0}^{\Delta T^s_{\max}} \sum_{\Delta T^a=0}^{\Delta T^a_{\max}} \int_a \mathcal{P}(\Delta T^s, \Delta T^a)\mathcal{P}(s'|s,a)\pi(a|I_{t-(\Delta T^s+\Delta T^a)})da \\
&= \sum_{(\Delta T^a+\Delta T^s)=0}^{\Delta T^a_{\max}+\Delta T^s_{\max}} \int_a \mathcal{P}(\Delta T^s+\Delta T^a)\mathcal{P}(s'|s,a)\pi(a|I_{t-(\Delta T^s+\Delta T^a)})da.
\end{aligned}
$$

From the outcome, we observe that the delay between decision-making and environmental response $\Delta T^a$ and the delay between environmental output and decision-making $\Delta T^s$ have the same effect on state transition, and what matters is merely $\Delta T^s + \Delta T^a$. Therefore, we can simply model the MDP with delay problem with only observation delay $\Delta T = \Delta T^s + \Delta T^a$ as in DOMDP.

$\square$

### D.2 Theorem 4.1 Derivation

*Proof.* To show the Markovian property when considering the augmented state ($\Delta T$ is max delay)

$$\bar{s}_t = (\tilde{s}_t, a_{t-\Delta T:t-1}),$$

we need to show the transition probability function $\bar{\mathcal{P}}(\bar{s}_{t+1}|\bar{s}_t, a_t)$ is independent of the time step $t$ (Markovian):

$$\bar{\mathcal{P}}(\bar{s}_{t+1}|\bar{s}_t, a_t)$$
$$=\tilde{\mathcal{P}}(\tilde{s}_{t+1}, a_{t-\Delta T+1:t}|\tilde{s}_t, a_{t-\Delta T:t})$$
$$= \sum_{\delta T'=0}^{\Delta T} \sum_{\delta T=0}^{\Delta T} \mathcal{P}(\delta T')\mathcal{P}(\delta T)\mathcal{P}(s_{t+1-\delta T'}, a_{t-\Delta T+1:t-1}|s_{t-\delta T}, a_{t-\Delta T:t})$$
$$= \sum_{\delta T'=0}^{\Delta T} \sum_{\delta T=0}^{\Delta T} \mathcal{P}(\delta T')\mathcal{P}(\delta T)\mathcal{P}(s_{t+1-\delta T'}|s_{t-\delta T}, a_{t-\Delta T:t}),$$

where $\mathcal{P}(\delta T)$ is the probability of delay for $\delta T$ steps for $\tilde{s}_t$, and $\mathcal{P}(\delta T')$ is the probability of delay for $\delta T'$ steps for $\tilde{s}_{t+1}$. Since $P(\delta T')$ and $P(\delta T)$ do not depends on $t$ according to the definition of DOMDP, we only need to prove that the last term $\mathcal{P}(s_{t+1-\delta T'}|s_{t-\delta T}, a_{t-\Delta T:t})$ is also independent of $t$ (with $0 \leq \delta T, \delta T' \leq \Delta T$). Let us discuss case by case:

When $\delta T' > \delta T + 1$, we have $\mathcal{P}(s_{t+1-\delta T'}|s_{t-\delta T}, a_{t-\Delta T:t}) = 0$ because $t + 1 - \delta T' < t - \delta T$ and the previous state should not depend on the future one.

When $\delta T' = \delta T + 1$, we have $\mathcal{P}(s_{t+1-\delta T'}|s_{t-\delta T}, a_{t-\Delta T:t}) = \mathcal{P}(s_{t-\delta T}|s_{t-\delta T}) = 1$.

When $\delta T' = \delta T$, we have $\mathcal{P}(s_{t+1-\delta T'}|s_{t-\delta T}, a_{t-\Delta T:t}) = \mathcal{P}(s_{t-\delta T+1}|s_{t-\delta T}, a_{t-\delta T})$, which is Markovian since it is the state transition function of the original MDP.

When $\delta T' = \delta T - 1$, we have

$$\mathcal{P}(s_{t+1-\delta T'}|s_{t-\delta T}, a_{t-\Delta T:t})$$
$$=\mathcal{P}(s_{t-\delta T+2}|s_{t-\delta T}, a_{t-\delta T}, a_{t-\delta T+1})$$
$$=\int_{s_{t-\delta T+1}} P(s_{t-\delta T+2}|s_{t-\delta T+1}, a_{t-\delta T+1})P(s_{t-\delta T+1}|s_{t-\delta T}, a_{t-\delta T})ds_{t-\delta T+1},$$

where each of the terms is the state transition function of the original MDP, thus $\mathcal{P}(s_{t+1-\delta T'}|s_{t-\delta T}, a_{t-\Delta T:t})$ is Markovian.

Similarly, when $\delta T' < \delta T - 1$, $\mathcal{P}(s_{t+1-\delta T'}|s_{t-\delta T}, a_{t-\Delta T:t})$ can be written as accumulated integration of state transitions of the original MDP at the different steps, therefore remaining independent of $t$.

So far, we have shown that $P(s_{t+1-\delta T'}|s_{t-\delta T}, a_{t-\Delta T:t})$ is Markovian and thus $\bar{\mathcal{P}}(\bar{s}_{t+1}|\bar{s}_t, a_t)$ is also Markovian. $\qquad\square$

Note that a similar theorem was also derived in (Chen et al., 2021). However, they considered fixed delay step while we consider the more general case of unfixed delay.

## E  IMPLEMENTATION DETAILS

### E.1  ENVIRONMENT SETUP

**(1) Fixed Delay:** This category includes four fundamental Mujoco environments: Walker2D, Ant, HalfCheetah, and Hopper. These environments are generally well-handled by existing methods, avoiding issues such as local optima (e.g., in Reacher) or insurmountable challenges (e.g., in HumanStandup). Thus, they provide suitable baselines for evaluating the impact of domain delay. We modify these environments to return the state from $\Delta T$ steps in the past, thereby simulating a fixed delay. For implementation, we use a FIFO queue with length as $\Delta T$ and update the queue with the new state always return with the one at the end of the queue to the agent.

**(2) Unfixed Delay:** This simulation reflects real-world scenarios with random delays (like network latency). We implement it using a FIFO queue of length $\Delta T$ that updates with new states. Each step samples from the queue to give the agent its observation. A straightforward normal sampling isn't used as real-world observations preserve order. Instead, we use a unique sampling strategy, where

we sample from $\{0, 1, 2\}$ shifts compared to the previous step. This maintains order and causes delays to range from 0 to $\Delta T$ with an average delay proportional to $\Delta T$.

**(3) Large State Space:** The dimensionality of the state space can significantly influence the feasibility of recovering the oracle state from delays. To study this, we utilized the Humanoid environment in Mujoco, which has a state dimension of 376, far larger than that of Ant (27), HalfCheetah (17), Hopper (11), or Walker (17).

**(4) Probabilistic:** To approximate the inherent uncertainties of real-world control systems, we applied a probabilistic adaptation to the deterministic Mujoco environments (Todorov et al., 2012). We employed two methods to introduce randomness: (1) Noise action, where Gaussian noise, $\mathcal{N}(0, 0.3)$, is added to the input action implemented by the environment; and (2) Sticky action, where there's a 30% chance that the environment will execute the previous action instead of the current input.

### E.2 Implementation of Baselines from Other Researches

This study specifically examines the solving of problem of delayed signal, a characteristic of the environment. Consequently, we modify only the environmental aspects while maintaining the integrity of the baseline algorithms. To ensure accuracy, we strictly utilize the original authors' source code for baseline implementations, altering only the environmental parameters to prevent implementation errors. Additionally, considering that all the baselines (Han et al., 2020; Ni et al., 2022; Chen et al., 2021) have previously been tested in the Mujoco environment as reported in their papers, it is reasonable for us to employ the default parameters established in those studies. This consistency in environmental settings enables a more direct and valid comparison of our findings with existing research.

### E.3 Network Architecture

In this study, we maintain a consistent architecture design across all investigated variants, with a focus on training paradigms and frameworks rather than network specifics. For both actor and critic networks in standard algorithms like SAC, DDPG, and TD3, we use a two-layer MLP with 256 features per layer, employing tanh as the activation function. In the case of RNN networks, we employ two layers of recurrent cells, also with 256 hidden features, to align the number of parameters. For the transformer structure, we use a single-head attention mechanism where key and value share the same tensor, and the query has a separate tensor, each with a feature size of 256. This attention block is applied twice, mirroring the two-layer MLP approach. For prediction-based methods, including those involving encoding, the process involves first encoding inputs into a 256-feature space using a two-layer MLP, followed by another two-layer MLP for making predictions. This uniform architecture across various models allows for a focused analysis on training methodologies, ensuring that observed performance differences are attributable to the training paradigm rather than structural variations.

### E.4 Hyperparameters

For a fair and consistent evaluation, we adhere to default hyperparameters as outlined in foundational studies for each algorithm. In the case of SAC, we follow parameters from Haarnoja et al. (2018b). Our approach for technique-specific parameters begins with standard settings, followed by adjustments within a practical range. This includes tuning the prediction loss weight in methods involving prediction and encoding, testing values in the set {0.005, 0.01, 0.05, 0.1}. Additionally, we explore auto KL weight tuning, setting target KL loss ranges, and sweeping through values {5, 20, 50, 200}. For input format consistency, especially in RNN-based models, we use 32 sequences of length 64. The memory buffer size is matched to the number of environment steps.

In scenarios involving multiple parameters, we average performance across each {algorithm, environment, parameter} combination, subsequently identifying parameters that yield the best average performance for each {algorithm, environment} pair. Then, we calculate the average performance of all the environments. This method ensures that performance is not compromised by suboptimal hyperparameter selection.

### E.5 Delyed Implementation

We developed an accessible plug-in environment wrapper for delayed environments, utilizing the `gym.Wrapper` from the OpenAI Gymnasium library (Brockman et al., 2016). This wrapper seamlessly integrates with pre-existing environments. In the *fixed delay* setting, the environment consistently returns observations from $T$ time steps prior. For the more complex and practical *unfixed delay* setting, the delay can range from $0$ to $T$ timesteps. To make it more realistic, we imposed a strict condition ensuring that the incoming observation does not arrive later than previous observations. Furthermore, our implementation accommodates action delay, in accordance with the equivalence established in Sec. 2.2.

### E.6 Code Implementation

We designed a modular and configurable implementation of all methods in this paper with seamless compatibility with popular state-of-the-art offline DRL algorithms including DDPG, TD3, SAC. Through which all the designs mentioned in this paper can be easily load or unload for efficient application and further studies. For the hyperparameters of the algorithm, we align them with the values in the original paper where each algorithm was proposed.

### E.7 Additional Details

Our performance analysis is derived from eight independent runs, ensuring statistical reliability. For the "Fixed/Basic" and "Unfixed" environments, evaluations were conducted after 2 million environmental steps, while assessments in other environments were completed after 1 million steps. This approach is based on the understanding that varying environments require different levels of interaction for effective learning.

## F Supplementary Experimental Results

### F.1 Is Representation Learning a Bottleneck?

In this section, we endeavor to design an experiment to ascertain whether the failure of certain baseline models can be attributed to deficiencies in representation learning. Specifically, we examine the Variational Recurrent Model (VRM) as proposed by Han et al. (2020) as a case study. VRM is a method for solving POMDP problems by using a variational recurrent neural network (Chung et al., 2015) which predict next observation from contextual observation and actions. Table 1 shows VRM's dramatic suffering from observation delay. To investigate whether this failure is due to unsuccessful representation learning. We present the representation learning loss (negative evidence lower bound) of the variational recurrent neural network of VRM in each tasks (Table 2). The results reflect no significant difference of the representation learning loss for delay steps 0, 1, 2 and 4. However, VRM's performance (Table 1) with delay=4 is only around one-third of the case with delay=0. Thus, it appears implausible that the difficulties VRM encounters in handling delayed tasks are caused by ineffective representation learning. This suggests that other factors may be contributing to the observed performance degradation.

| Delay | Walker2d | HalfCheetah | Hopper | Ant | Average |
|:---:|:---:|:---:|:---:|:---:|:---:|
| **0** | $-1.87 \pm 0.31$ | $-1.60 \pm 0.15$ | $-2.50 \pm 0.07$ | $-1.68 \pm 0.21$ | -1.91 |
| **1** | $-1.90 \pm 0.20$ | $-1.74 \pm 0.10$ | $-2.54 \pm 0.05$ | $-1.99 \pm 0.43$ | -2.04 |
| **2** | $-2.25 \pm 0.24$ | $-1.70 \pm 0.20$ | $-2.62 \pm 0.10$ | $-1.89 \pm 0.21$ | -2.10 |
| **4** | $-2.26 \pm 0.06$ | $-1.65 \pm 0.22$ | $-2.49 \pm 0.09$ | $-1.81 \pm 0.24$ | -2.05 |
| **8** | $-1.92 \pm 0.18$ | $-1.57 \pm 0.08$ | $-1.81 \pm 0.08$ | $-1.67 \pm 0.20$ | -1.74 |
| **12** | $-1.86 \pm 0.14$ | $-1.29 \pm 0.20$ | $-1.56 \pm 0.14$ | $-1.31 \pm 0.10$ | -1.50 |

Table 2: Representation learning loss (lower the better) of VRM in the end of learning for various environments with different delay steps (fixed delay). Data are mean $\pm$ standard deviation.

## F.2 DOES REDUNDANT HISTORICAL INFORMATION HAMPER THE PERFORMANCE?

In addressing concerns about the potential redundancy of historical information in our model, we conducted experiments to determine the optimal quantity of historical data for enhancing performance. Specifically, we varied the number of past actions ($H$) considered by the model and assessed its impact under fixed delay conditions ($\Delta t = \{0, 1, 4\}$).

Our findings, detailed in Tab. 3, reveal that excluding past actions ($H = 0$) results in suboptimal performance. For instance, with $\Delta t = 4$, the absence of historical actions led to a low performance score of 47.6. Conversely, incorporating a moderate history ($H$) significantly improves performance, as shown by a score of 84.3 for $H = 2$. However, excessively large histories ($H \geq 8$) negatively impact performance, evident from a decrease to 55.0 for $H = 12$. This trend is consistent across different $\Delta t$ values, indicating that a balanced approach to historical data is crucial.

Our experiments employed both "Delay Reconciled Training for Critic" and "State Augmentation - MLP" methods, maintaining consistency across all variables except $H$. The results also shed light on the limited success of RNN structures in our experiments. RNNs inherently process extensive historical information, leading to redundancy, particularly for state and action histories beyond the $\Delta t$ window. This aligns with our theoretical finding (Theorem 4.1) that maintaining the Markovian property requires only historical actions within $\Delta t$.

Table 3: Impact of Historical Information ($H$) on Model Performance under Different Delays ($\Delta t$). The under-performing methods are marked with ↓.

| Historical Timesteps $H$ ↓ | # Delay $\Delta t$ | | |
|---|---|---|---|
| | 0 | 1 | 4 |
| $H = 0$ | 97.7±24.0 | 87.8±22.3 | 47.6±20.3↓ |
| $H = 1$ | 104.5±18.4 | 100.2±16.5 | 77.7±15.4 |
| $H = 2$ | 98.8±23.4 | 93.7±18.9 | 84.3±16.3 |
| $H = 4$ | 86.8±18.6 | 81.6±23.4 | 80.3±17.8 |
| $H = 8$ | 68.3±12.5↓ | 73.0±13.6↓ | 58.6±15.5 |
| $H = 12$ | 62.6±14.3↓ | 60.9±12.8↓ | 55.0±8.3↓ |

## F.3 COMPARING FIXED AND UNFIXED DELAY

Here, we report additional results of algorithms on fixed and unfixed delay in Tab. 7. As most algorithm performances are already reported in Tab. 1, we primarily focus on the comparison of Prediction and Encoding methods' variations. Our findings reveal that detached methods consistently outperform non-detached methods. Specifically, for prediction-based methods, the average performance increases from 68.0% to 83.6% under fixed delay, and 58.8% to 72.5% under unfixed delay. For encoding-based methods, we observe an average performance increase from 76.1% to 84.5% under fixed delay and 70.2% to 77.9% under unfixed delay. These results suggest that the detached design is preferable.

## F.4 LARGE OBSERVATION SPACE

The performance of various algorithms in environments with large observation spaces is presented in Tab. 5. Consistently, both Delay-Reconciled Training and State Augmentation - MLP yield positive performance enhancements. Notably, State Augmentation - MLP consistently offers the best performance. Conversely, the performances of Prediction[†] and Encoding[†] are substantially inferior, implying that explicitly introducing prediction supervision may adversely impact performance when the observaton space is large.

## F.5 PROBABILISTIC SETTINGS - GAUSSIAN MUJOCO

In this subsection, we explore the impact of probabilistic state transitions by introducing diagonal Gaussian noise $\mathcal{N}(0, \sigma^2)$ to the Mujoco environment. This approach tests our algorithm's robustness under varying degrees of uncertainty.

Table 4: **Performance (%) of algorithms in fixed and unfixed delay environments (additional comparation of detach and non-detach)**. The best performing methods of prediction and encoding are highlighted in **bold**.

| Method | # Delayed Time Steps | | | | | | | | | | | Avg. | #≥4 |
|---|---|---|---|---|---|---|---|---|---|---|---|---|---|
| | 0 | 1 | | 2 | | 4 | | 8 | | 12 | | | |
| *Prediction* | | | | | | | | | | | | | |
| Prediction | 103.1 ±7.8 | 77.7 ±11.3 | 75.2 ±9.5 | 77.1 ±11.5 | 80.2 ±10.2 | 73.2 ±13.2 | 70.4 ±11.6 | 70.1 ±8.6 | 57.3 ±10.1 | 59.6 ±8.3 | 47.1 ±11.4 | 68.0 ±10.0 | 58.8 ±11.1 |
| Prediction $^\dagger$ | 102.1 ±9.7 | **101.5** **±13.9** | **96.1** **±12.2** | **100.5** **±14.4** | **99.0** **±12.5** | 92.2 ±16.2 | **88.6** **±14.4** | 85.6 ±10.8 | **71.0** **±12.8** | 73.0 ±10.3 | **58.1** **±14.1** | **83.6** **±12.5** | **72.5** **±13.8** |
| *Encoding* | | | | | | | | | | | | | |
| Encoding | 98.5 ±10.9 | 91.7 ±8.1 | 102.5 ±8.7 | 93.1 ±9.0 | 83.4 ±8.5 | 81.8 ±8.5 | 77.9 ±7.4 | 80.1 ±11.2 | 70.2 ±9.7 | 66.0 ±8.9 | 62.5 ±8.7 | 76.1 ±9.5 | 70.2 ±8.7 |
| Encoding $^\dagger$ | 97.6 ±12.1 | **101.4** **±9.0** | **113.0** **±9.6** | **103.2** **±10.0** | **92.0** **±9.5** | 90.4 ±9.5 | **86.6** **±8.2** | 89.9 ±12.4 | **77.9** **±10.8** | 73.3 ±9.9 | **69.2** **±9.7** | **84.5** **±10.6** | **77.9** **±9.6** |

Table 5: Performance (%) of algorithms in environment with large observation space. The top-performing methods are highlighted in **bold**.

| Method | # Delayed Time Steps | | | | | |
|---|---|---|---|---|---|---|
| | 0 | 1 | 2 | 4 | 8 | 12 |
| Vanilla SAC | 24.7±13.6 | 11.8±1.9 | 12.3±2.8 | 2.5±1.2 | 2.6±1.7 | 2.9±1.8 |
| Delay-Reconciled Training | 23.9±6.4 | 26.6±7.2 | 21.8±8.1 | 17.3±4.7 | 13.2±2.5 | 14.6±3.1 |
| State Augmentation - MLP | **83.8±22.9** | **103.4±11.5** | **94.6±11.9** | **102.7±6.4** | **101.3±7.8** | **89.7±14.5** |
| State Augmentation - RNN | 48.3±30.8 | 67.7±39.4 | 47.8±38.6 | 47.5±43.7 | 53.6±45.8 | 55.4±42.6 |
| Encoding$^\dagger$ | 87.5±24.8 | 97.3±15.7 | 67.9±21.8 | 20.4±4.7 | 18.2±3.0 | 14.5±2.9 |
| Prediction$^\dagger$ | 87.7±32.6 | 22.3±6.2 | 16.6±3.7 | 18.5±3.9 | 16.9±2.4 | 14.4±2.9 |

Experiments were conducted with noise variances $\sigma = \{0.05, 0.1, 0.2, 0.4\}$, comparing our modified algorithm (Ours$^*$) against the SAC. Notably, at delay=0, Vanilla SAC and Ours$^*$ performances are equivalent.

The performance metrics are presented as "Value(Relative%)", The "Value" is consistent with the normalization in Tab. 1. The "Relative%" indicates the relative reward compared to the scenario when delay is 0. Key observations include:

- Vanilla SAC shows a marked performance decrease with delay, dropping to 23.78%, 26.67%, 32.39%, and 44.25% for $\sigma$ values of 0.05, 0.1, 0.2, and 0.4, respectively.
- Our method significantly enhances performance in delayed scenarios, with improvements from 23.78% to 79.57%, 26.67% to 83.14%, 32.39% to 92.25%, and 44.25% to 98.23% for the respective $\sigma$ values.

These results demonstrate the effectiveness of our algorithm in environments with probabilistic state transitions, underscoring its robustness and adaptability.

### F.6 Probabilistic Settings - Noisy and Sticky Actions

This section reveals additional numerical results for algorithms within probabilistic state transition environments. Performance measures for two probabilistic environment implementations, "sticky action" and "noisy action", are outlined in Tab. 7 and Tab. 8 respectively.

Notably, both the "noisy action" and "sticky action" settings intensify the environment's complexity, causing a performance decrease from 100.0% to 72.3% and 64.3%, respectively. For both the "noisy action" and "sticky action" environments, Delay-reconciled training and State Augmentation - MLP exhibit a consistent performance improvement over vanilla SAC in all settings where the delay exceeds one step. This suggests that our design is effective in probabilistic environments as well. However, prediction and encoding-based methods fail to perform satisfactorily within these settings.

Table 6: Performance (%) of algorithms in probabilistic (Gaussian State) environment. The top-performing methods are highlighted in **bold**.

| Scale of Randomness | Method | # Delayed Time Steps | | | | | |
|---|---|---|---|---|---|---|---|
| | | 0 | 1 | 2 | 4 | 8 | 12 |
| $\sigma = 0.05$ | Ours* | $32.8 \pm 7.4$ (100%) | $35.0 \pm 7.9$ (106.71%) | $32.9 \pm 5.7$ (100.30%) | $31.7 \pm 5.8$ (96.65%) | $27.1 \pm 7.2$ (82.62%) | $26.1 \pm 6.0$ (79.57%) |
| $\sigma = 0.05$ | Vanilla SAC | $32.8 \pm 7.4$ (100%) | $12.4 \pm 3.9$ (37.80%) | $11.0 \pm 1.9$ (33.54%) | $9.7 \pm 1.5$ (29.57%) | $8.1 \pm 1.1$ (24.70%) | $7.8 \pm 1.3$ (23.78%) |
| $\sigma = 0.1$ | Ours* | $25.5 \pm 2.9$ (100%) | $23.1 \pm 4.0$ (90.59%) | $24.3 \pm 5.3$ (95.29%) | $21.2 \pm 4.7$ (83.14%) | $21.9 \pm 6.8$ (85.88%) | $21.2 \pm 4.2$ (83.14%) |
| $\sigma = 0.1$ | Vanilla SAC | $25.5 \pm 2.9$ (100%) | $11.0 \pm 1.4$ (43.14%) | $10.4 \pm 1.7$ (40.78%) | $9.4 \pm 1.3$ (36.86%) | $7.4 \pm 1.2$ (29.02%) | $6.8 \pm 3.3$ (26.67%) |
| $\sigma = 0.2$ | Ours* | $14.2 \pm 1.2$ (100%) | $13.3 \pm 1.9$ (93.66%) | $13.3 \pm 1.8$ (93.66%) | $13.4 \pm 2.1$ (94.37%) | $11.8 \pm 2.3$ (83.10%) | $13.1 \pm 2.5$ (92.25%) |
| $\sigma = 0.2$ | Vanilla SAC | $14.2 \pm 1.2$ (100%) | $10.9 \pm 1.9$ (76.76%) | $10.1 \pm 1.1$ (71.13%) | $8.6 \pm 1.5$ (60.56%) | $8.0 \pm 1.2$ (56.34%) | $4.6 \pm 5.0$ (32.39%) |
| $\sigma = 0.4$ | Ours* | $11.3 \pm 1.7$ (100%) | $11.0 \pm 1.4$ (97.35%) | $10.0 \pm 2.3$ (88.50%) | $10.8 \pm 2.1$ (95.58%) | $10.7 \pm 1.7$ (94.69%) | $11.1 \pm 1.1$ (98.23%) |
| $\sigma = 0.4$ | Vanilla SAC | $11.3 \pm 1.7$ (100%) | $10.0 \pm 1.2$ (88.50%) | $10.1 \pm 1.2$ (89.38%) | $8.8 \pm 2.1$ (77.88%) | $8.0 \pm 1.8$ (70.80%) | $5.0 \pm 5.4$ (44.25%) |

Most notably, the prediction method's performance in a noisy action environment drastically drops from 72.8% to 9.9% with just a one-step delay. More importantly, prediction and encoding methods consistently underperform when compared to State Augmentation - MLP, implying that their application does not invariably yield positive performance effects on environments with probabilistic state transition.

Table 7: Performance (%) of algorithms in probabilistic (noise action) environment. The top-performing methods are highlighted in **bold**.

| Method | # Delayed Time Steps | | | | | |
|---|---|---|---|---|---|---|
| | 0 | 1 | 2 | 4 | 8 | 12 |
| Vanilla SAC | 72.3±12.4 | 48.1±14.7 | 38.6±12.9 | 14.5±9.4 | 3.2±3.5 | 0.7±3.8 |
| Delay-Reconciled Training | 74.7±9.1 | 66.6±9.2 | 59.8±8.6 | 48.3±8.3 | 25.4±6.9 | 14.2±4.7 |
| State Augmentation - MLP | **77.9±9.2** | **71.3±11.5** | **65.7±10.3** | **64.6±11.9** | **48.8±10.2** | **34.1±14.7** |
| State Augmentation - RNN | 65.4±15.6 | 60.7±11.4 | 54.1±10.5 | 40.9±10.3 | 20.3±8.7 | 12.5±5.2 |
| Prediction[†] | 72.8±18.3 | 9.9±4.8 | 20.4±10.2 | 24.7±8.1 | 17.6±3.2 | 5.3±1.1 |
| Encoding[†] | 35.7±19.8 | 41.2±19.4 | 43.8±12.7 | 39.5±11.6 | 25.9±5.7 | 25.4±1.9 |

Table 8: Performance (%) of algorithms in probabilistic (sticky action) environment. The top-performing methods are highlighted in **bold**.

| Method | # Delayed Time Steps | | | | | |
|---|---|---|---|---|---|---|
| | 0 | 1 | 2 | 4 | 8 | 12 |
| Vanilla SAC | 64.3±18.7 | 44.6±16.1 | 26.5±15.8 | 9.4±10.3 | 6.7±3.4 | 0.8±3.9 |
| Delay-Reconciled Training | 63.9±15.6 | **70.8±18.2** | **66.7±12.9** | 52.4±11.5 | 29.3±9.2 | 18.2±9.6 |
| State Augmentation - MLP | **68.7±11.2** | 62.4±13.5 | 62.6±18.3 | **57.8±21.9** | **38.9±13.6** | **32.3±11.5** |
| State Augmentation - RNN | 57.5±19.8 | 57.1±13.4 | 51.9±14.8 | 39.3±18.6 | 16.6±8.7 | 15.7±8.4 |
| Prediction[†] | 52.9±14.6 | 27.3±3.9 | 27.8±6.1 | 21.4±7.2 | 13.5±4.3 | 12.6±4.9 |
| Encoding[†] | 42.7±4.8 | 37.2±2.7 | 37.4±2.4 | 34.8±12.6 | 29.3±7.2 | 26.9±4.8 |

## F.7 COMPREHENSIVE ANALYSIS OF LEARNING CURVES

This section provides an in-depth analysis of the learning curves exhibited by various algorithms across distinct environments. Figures 8 through 15 illustrate the progression of learning in environments characterized by consistent, fixed delays. In contrast, figures 16 through 23 present the learning trajectories within environments where delays are variable and unfixed. The results depicted here are intended to furnish a comprehensive benchmark for subsequent research in this domain. Additionally, we highlight several key insights derived from our analysis.

1. Even a 2-step delay can significantly impair the learning efficiency of standard algorithms, leading to nearly flat learning curves.

2. The impact of delay varies dramatically across different environments. For example, the Hopper environment is relatively less affected by delay, experiencing an overall performance drop of around 10% as illustrated in Fig. 13 and 21. On the other hand, the HalfCheetah environment sees a substantial performance decline of approximately 60% with a 12-step delay.

3. The presence of delay affects both the speed of learning and optimal performance. As the number of delay steps increases, the learning curve usually shows a steady rise in most cases, indicating the gradual adaptation of the learning process to the delay.

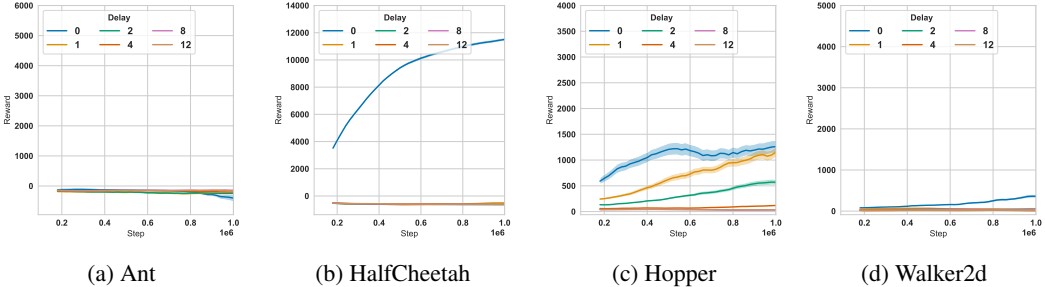

(a) Ant  (b) HalfCheetah  (c) Hopper  (d) Walker2d

Figure 8: Learning curve of algorithms on environment with fixed delay: vanilla DDPG.

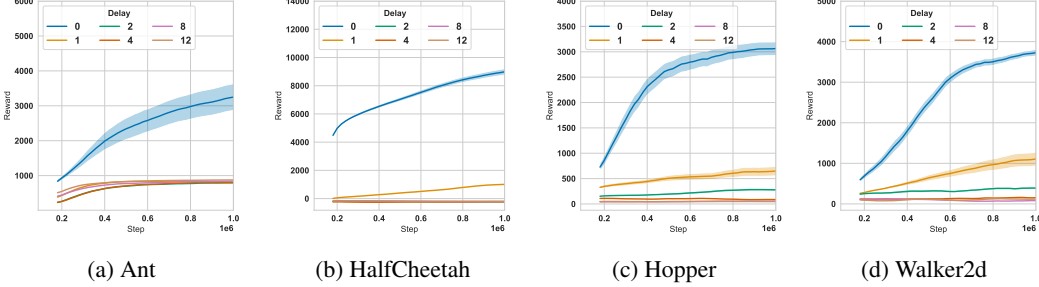

(a) Ant  (b) HalfCheetah  (c) Hopper  (d) Walker2d

Figure 9: Learning curve of algorithms on environment with fixed delay: vanilla TD3.

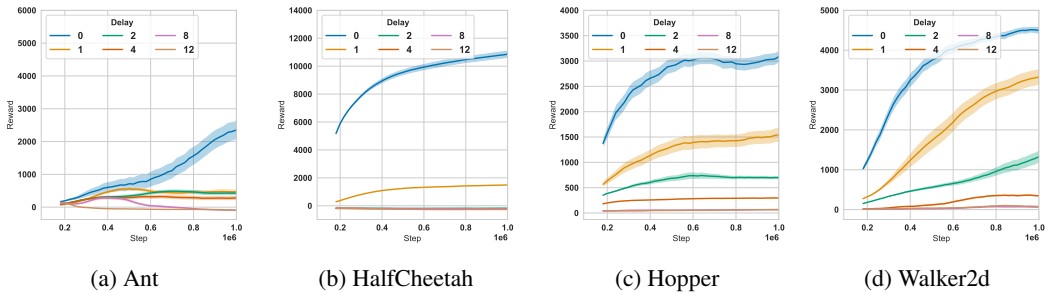

Figure 10: Learning curve of algorithms on environment with fixed delay: vanilla SAC.

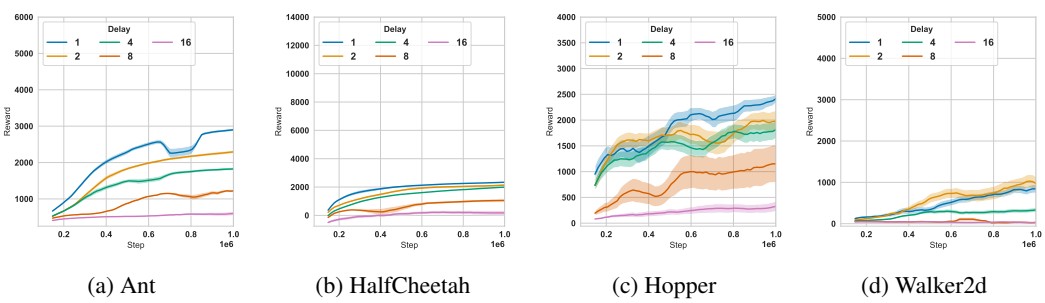

Figure 11: Learning curve of algorithms on environment with fixed delay: RNN Strong.

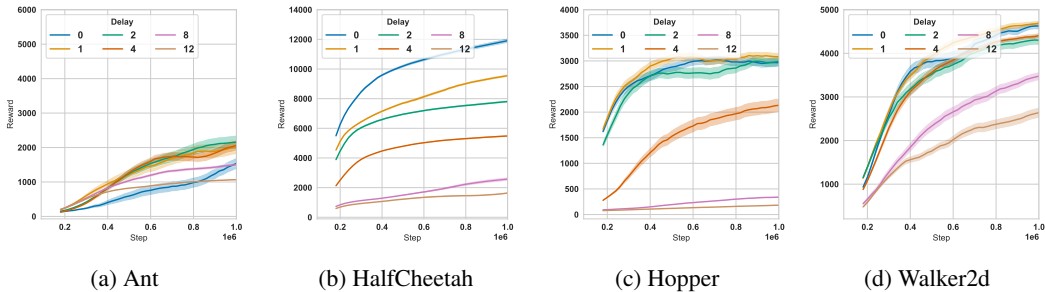

Figure 12: Learning curve of algorithms on environment with fixed delay: delay-reconciled critic training.

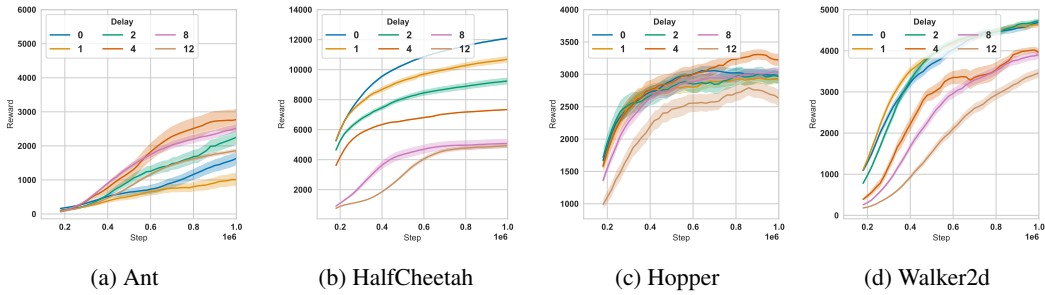

Figure 13: Learning curve of algorithms on environment with fixed delay: state augmentation - MLP.

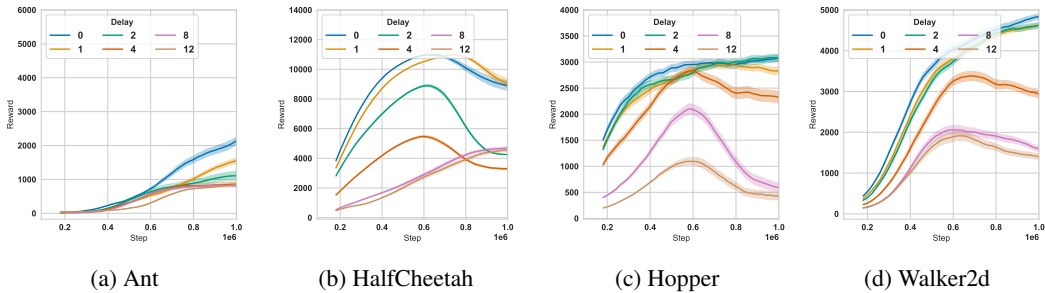

Figure 14: Learning curve of algorithms on environment with fixed delay: state augmentation - RNN.

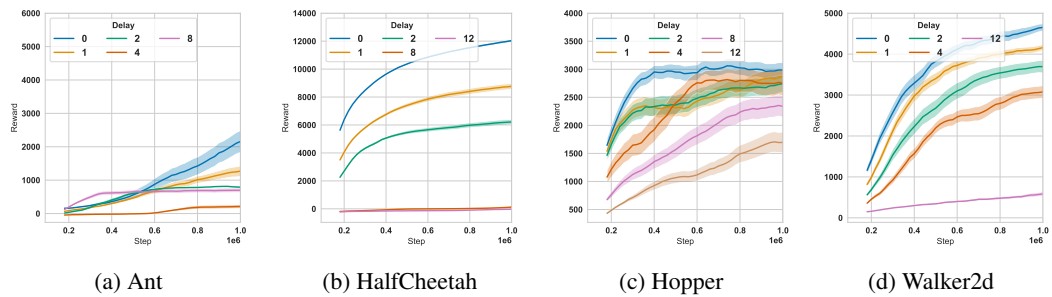

Figure 15: Learning curve of algorithms on environment with fixed delay: symmetric - MLP.

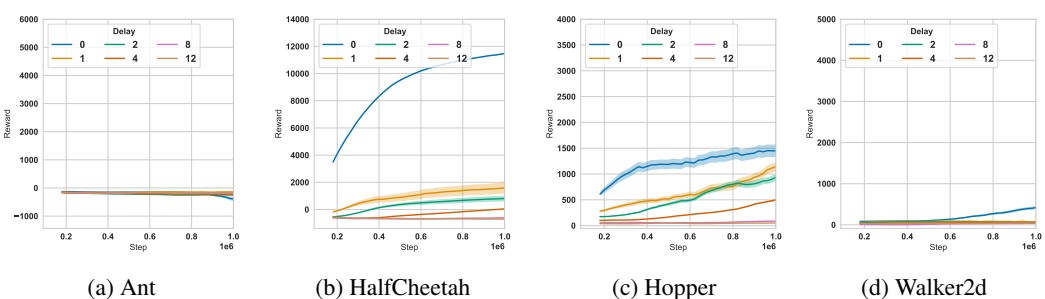

Figure 16: Learning curve of algorithms on environment with unfixed delay: vanilla DDPG.

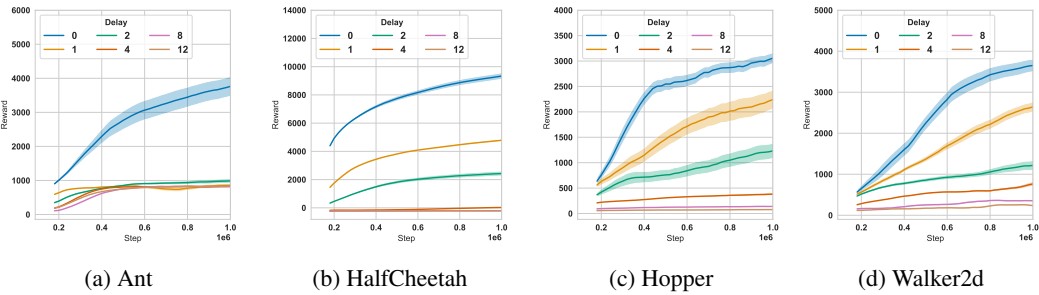

Figure 17: Learning curve of algorithms on environment with unfixed delay: vanilla TD3.

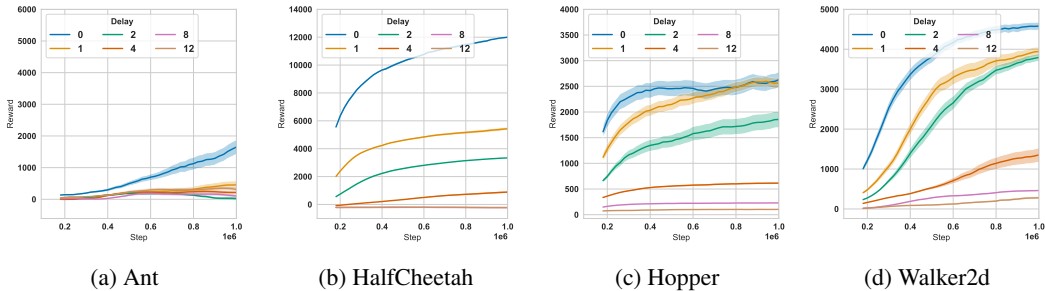

Figure 18: Learning curve of algorithms on environment with unfixed delay: vanilla SAC.

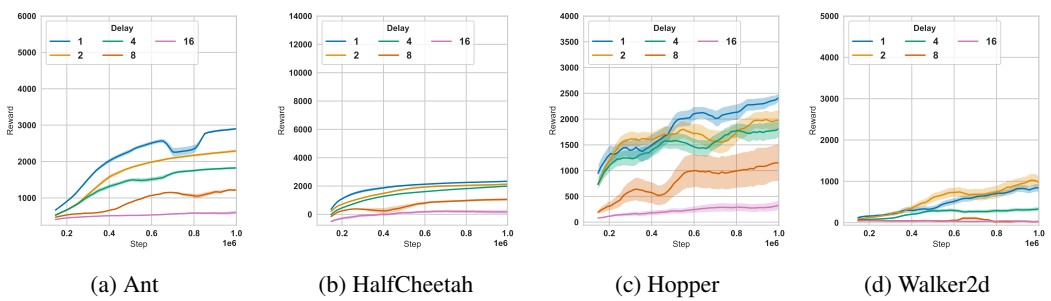

Figure 19: Learning curve of algorithms on environment with unfixed delay: RNN Strong.

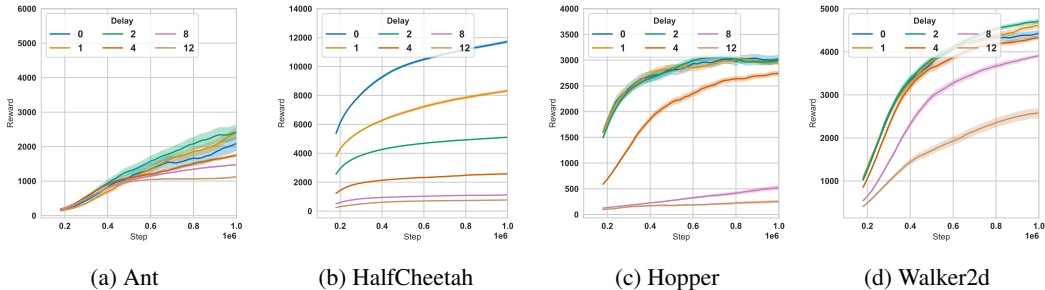

Figure 20: Learning curve of algorithms on environment with unfixed delay: delay-reconciled critic training.

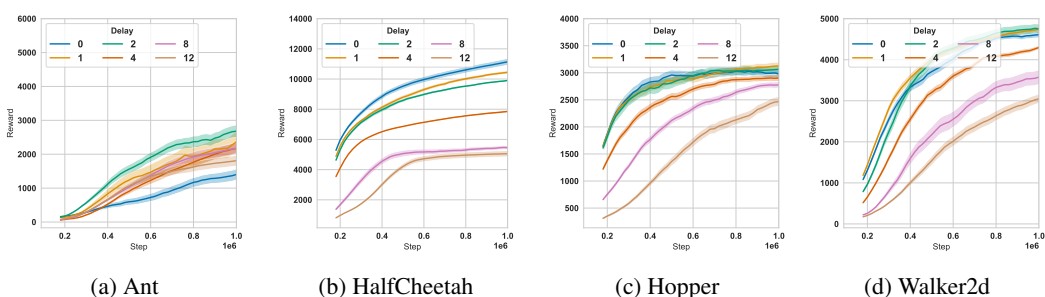

Figure 21: Learning curve of algorithms on environment with unfixed delay: state augmentation - MLP.

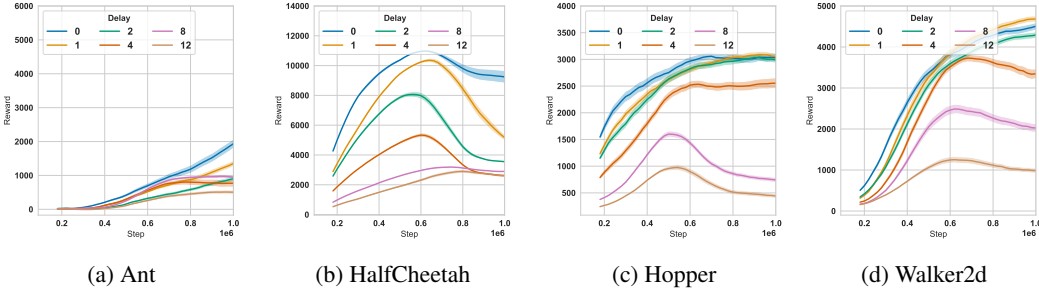

|   |   |   |   |
|---|---|---|---|
| (a) Ant | (b) HalfCheetah | (c) Hopper | (d) Walker2d |

Figure 22: Learning curve of algorithms on environment with unfixed delay: state augmentation - RNN.

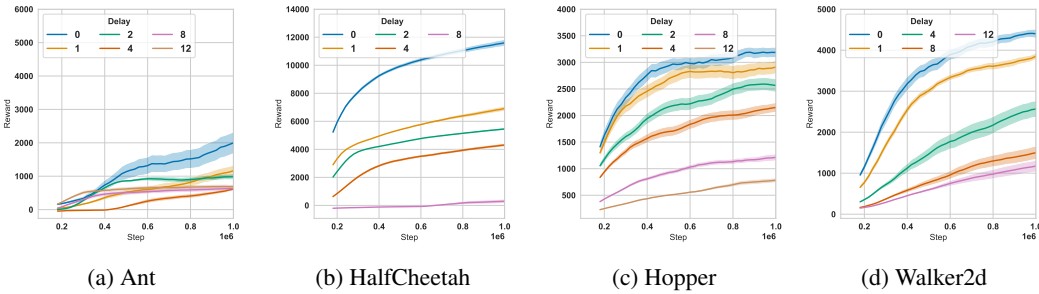

|   |   |   |   |
|---|---|---|---|
| (a) Ant | (b) HalfCheetah | (c) Hopper | (d) Walker2d |

Figure 23: Learning curve of algorithms on the environment with unfixed delay: symmetric - MLP.

