# OpenReview forum: "Addressing Signal Delay in Deep Reinforcement Learning"
_ICLR.cc/2024/Conference — ICLR 2024 spotlight_

### Official Review · Reviewer_6hPB · 2023-10-29

**Soundness:** 2 fair
**Presentation:** 2 fair
**Contribution:** 1 poor
**Rating:** 3
**Confidence:** 4

**Summary:**

This paper focuses on the problem of sequential decision making with signal delay.

That means observations on the current state do not arrive immediately after an action is taken, but arrive several steps after.

This delay in signal effectively makes the problem a POMDP, even when the agent always observes the underlying environment state.

Tthe authors propose the formalization for such setting, called delayed osbervation MDP, and show that in fact it is a special case of POMDP.

While we can potentially see this kind of problem as a POMDP and use standard DRL algorithms for POMDPs, the authors argue and show that these algorithms struggle in such setting.

As the main contribution of this paper, several techniques are proposed to improve the performance of DRL algorithms, in particular, actor critic methods, in this setting.

These techniques mainly rely on using priviledged information, e.g., access to the true current state, during offline training to improve the learning of actor and critic.

Specifically, for the critic, the authors propose to condition the critic on the true current state instead of the history of observations or the current observation which might be the true state a few steps earlier during training. Because the critic is not used during inference or deployment, this approach is possible as long as we have access to the true states during training.
For the actor, the authors propose to add actions taken in the past to the inputs of the actor as this will provide useful information to help the actor to predict the true current state.

Experiments are performed to test the effectiveness of these proposed techniques. To do so, the authors take SAC, a strong model-free baseline, and build upon it with the proposed techniques. Results show that these techniques are indeed generally useful, especially when the delay is longer.

**Strengths:**

This is an interesting paper.

It looks at a problem that is somewhat ignored in the literature, indentifies the failure of existing algorithms, and proposes several techniques to improve the baselines, which are shown to be effective.

Many baselines are considered and compared in the experiments.

**Weaknesses:**

My biggest concern on this paper is that many of the proposed techniques, if not all, are not new, and the paper misses a large body of closely relevant works. In Section 4.1, the authors propose to condition the critic on the true current state (without delay) to improve learning. The idea of using priviledged information during training by conditioning the critic on the ground-truth state in actor critic methods has been widely explored by both the POMDP and MARL communities. For example, see [1] and [2] for the former case and [3] for the latter case. Section 4.1 basically describes a trivial application of this idea to DOMDPs, which are special cases of POMDPs. Furthermore, the authors mention that the efficay of such asymmetric architecture has been demonstrated by past previous studies. However, in [2], it has been shown that conditioning the critic (only) on the true state is fundamentally questionable as it leads to biased gradients.

Then in Section 4.2, the authors propose to give the actor past actions to help it infer the state at the current step. I don't understand why is this not done by default. In my understanding, DOMDPs are POMDPs and in POMDPs, past actions and observations should always be given to the policy for optimal control. I don't see how this is an innovation.

The method that is proposed in Section 4.3 has also been explored for the general POMDP settings. For example, in [4,5,6], people have explored auxiliary losses (to predict the next observation, state, reward) for improving the representation learning of RNNs when learning optimal policies in POMDPs.

In the beginning of Section 3, baseline DRL methods are tested in delay signal settings and they are demonstrated to struggle in such a setting. Reasons are guessed for why they fail. However, I do not think that the failure of these baselines methods are studied well enough. I would like to see more ablation study on understanding how exactly they fail, which I believe will greatly help readers understand in what way DOMDPs are different to POMDPs and what are their unique challenges. Is it really due to the signal delay? Or like in general POMDPs, the RNNs cannot come up with good representations for decision making? For example, if these methods struggle to learn representations that support effective decision making, then one can test the accuracy of using these learned representations to predict the true current state. There are possible causes mentioned in paper but I find them rather vague and are not enough to motivate the proposed algorithmic changes.

I also find lots of experimental details are missing from the paper, such as the architectures of networks, and the hyperparameters. Also, for the baselines, such as DDPG, TD3, SAC, are they also RNN based? It would be strange to use a non-recurrent baseline in a POMDP environment. And are the past actions passed to the actors in these baseline methods? Regarding hyperparameters, it would also be useful to know how are the hyperparameters tuned for each baseline and the proposed method to ensure fair comparison.

[1] Baisero, Andrea, Brett Daley, and Christopher Amato. "Asymmetric DQN for partially observable reinforcement learning." Uncertainty in Artificial Intelligence. PMLR, 2022.

[2] Baisero, Andrea, and Christopher Amato. "Unbiased Asymmetric Reinforcement Learning under Partial Observability." Proceedings of the 21st International Conference on Autonomous Agents and Multiagent Systems. 2022.

[3] Foerster, Jakob, et al. "Counterfactual multi-agent policy gradients." Proceedings of the AAAI conference on artificial intelligence. Vol. 32. No. 1. 2018.

[4] Igl, Maximilian, et al. "Deep variational reinforcement learning for POMDPs." International Conference on Machine Learning. PMLR, 2018.

[5] Subramanian, Jayakumar, et al. "Approximate information state for approximate planning and reinforcement learning in partially observed systems." The Journal of Machine Learning Research 23.1 (2022): 483-565.

[6] Lambrechts, Gaspard, Adrien Bolland, and Damien Ernst. "Informed POMDP: Leveraging Additional Information in Model-Based RL." arXiv preprint arXiv:2306.11488 (2023).

**Questions:**

In Figure 3, I find it weird to argue that baseline methods are failing by saying that there is a significant drop in performance from no delay to one-step delay. With different delay steps, we essentially have different POMDPs, which may inherently have different difficulty levels and different optimal performance. As such, the drop in performance I think is not sufficient to say the baseline methods are failing. However, a comparison to the optimal performance or performance from a better method would work.

minor:
- in Figure 3, colors correspond to delay lengths different in (d) than in other subplots. this can be confusing for readers.

---

> ### Author Response · Authors · 2023-11-23
> **Part 1**
>
> Thank you for your insightful comments and for raising the concern regarding our paper. We appreciate the opportunity to clarify our contribution in this regard.
>
> > “My biggest concern on this paper is that many of the proposed techniques, if not all, are not new, and the paper misses a large body of closely relevant works.
> >
>
> We apologize if our paper gave the impression that these techniques were newly proposed by us. We recognize the importance of acknowledging previous work and have updated to properly reference earlier studies where these techniques were first introduced and applied (see Sec.4.1 last paragraph, Sec.4.3 first paragraph).
>
> Our main contribution is not in creating brand new RL algorithms, but in systematically investigate effective framework for RL with delays, which can be applied to solve DOMPDs as a plug-in.
>
> For your concerns about every  technique you mentioned below, we are not simply use it, but either provide theoretical guaranteed effective modifications or comprehensive study of possible variants. We believe our work our work contribute to the research in the field of singal delay in RL, providing valuable insights and practical solutions.
>
> In the following, we are going to address your detailed concerns about these techniques one by one to explain how we contribute to each one.
>
> > In Section 4.1, the authors propose to condition the critic on the true current state (without delay) to improve learning. The idea of using priviledged information during training by conditioning the critic on the ground-truth state in actor critic methods has been widely explored by both the POMDP and MARL communities. For example, see [1] and [2] for the former case and [3] for the latter case. Section 4.1 basically describes a trivial application of this idea to DOMDPs, which are special cases of POMDPs.
> >
>
> It is true that using “ground-truth state” is a widely explored We address the following facts:
>
> 1. **There has been extensive studies about delayed RL while they all miss to use this “ground-truth state” technique.** We are the first to deploy it in delayed RL and the show that the performance improvement is significant. The techniques itself may be not novel but investigating it and reveals the robustness is an important step for Delay DRL study. Investigating and reveals this comprehensively is of a small part of our investigation of techniques which is potential to solve delayed RL.
> 2. **Existing SOTA POMDP methods can not solve delayed RL well.** We has conduct experiment with “RNN Strong” and VRM, both are SOTA methods in POMDP studies. But their performance improvement are both marginal. So, simply treating DOMDP as a special case of POMDP and using existing methods can not be a solve the problem perfectly.
>
> > Furthermore, the authors mention that the efficay of such asymmetric architecture has been demonstrated by past previous studies. However, in [2], it has been shown that conditioning the critic (only) on the true state is fundamentally questionable as it leads to biased gradients.”
> >
>
> This is a great point. Conditioning the critic (only) on the true state is asymmetric and could lead to bad performance.
>
> To address this, in our paper, we not only implemented the "Oracle Guiding Critic-MLP" but also conducted an ablation study with the "Symmetric-MLP" approach, using observations and historical actions (see Tab.1, final paragraph of Section 4.1).
>
> |  | Input-Critic | Input-Actor |
> | --- | --- | --- |
> | Oracle Guiding Critic-MLP | true-state $s_t$ | delayed state $s_t$ + historical actions $a_{t-\Delta T:t-1}$ |
> | Symmetric-MLP | delayed state $s_t$ + historical actions $a_{t-\Delta T:t-1}$ | delayed state $s_t$ + historical actions $a_{t-\Delta T:t-1}$ |
>
> We recognize there's a trade-off where the oracle critic can provide valuable information but also introduce bias. In this way, experimental investigation under delayed RL becomes crucial.
>
> |  | 0 | 1 | 2 | 4                                                                                           | 8 | 12 | Avg. |
> | --- | --- | --- | --- | --- | --- | --- | --- |
> | Oracle-Critic -MLP | 100.9±11.2 | 92.8±7.1 | 93.4±8.1 | 78.0±9.2 | 85.0±9.9 | 64.7±8.6 | 75.9±9.2 |
> | Symmetric-MLP | 106.3±9.4 | 75.1±9.1 | 63.7±9.7 | 51.4±11.7 | 37.5±13.2 | 27.0±12.0 | 38.6±12.2 |
>
> Our experimental results, as shown in the table below, indicate that the benefits of the information provided by the oracle critic generally outweigh the potential bias. We can see that “Oracle-Critic -MLP” consistently outperform the “Symmetric-MLP”, and the performance gap is not trivial, which is 38.6 v.s. 75.9 on average.
>
> These findings are consistent and we believe they will be valuable for future applications and research in this area.
>
> We recognize that our paper's initial presentation is not clear. To address this, we've revised Section 4.1's last paragraph for better accuracy and included relevant paper you mentioned.

---

> ### Author Response · Authors · 2023-11-23
> **Part 2**
>
> > “Then in Section 4.2, the authors propose to give the actor past actions to help it infer the state at the current step. I don't understand why is this not done by default. In my understanding, DOMDPs are POMDPs and in POMDPs, past actions and observations should always be given to the policy for optimal control. I don't see how this is an innovation.”
> >
>
> We agree that using observation and action history is did a common practice in POMDPs. But we want to address the following facts:
>
> 1. **The “common practices” can not solve Delayed RL well:** Although using historical observations and actions is a standard approach in POMDPs, our findings suggest that this strategy is less effective in DOMDPs. For instance, as demonstrated in Tab.1, the state-of-the-art POMDP method 'RNN Strong', which adheres to this common practice, using both action and state history, significantly underperforms compared to our proposed method. This discrepancy indicates that the “common practices” does not fully address the challenges posed by DOMDPs.
> 2. **We show that Historical Actions and only historical actions in the delayed window are useful:** Our research shows that not all historical information is helpful. Instead, only the actions within the delay window is useful. This is a departure from the “common practices” of incorporating all historical data. More specifically, We show that theoretically (Theorem 4.1):
>     - Historical states do not improve to performance in DOMDPs.
>     - Only actions within the delay window are helpful.
>
>     This selective incorporation of historical data not only save time and space cost. More importantly, it leads to a notable improvement in performance. We provide experimental evidence to support this claim, which also addresses your subsequent question about the RNN model so we defer there.
>
>
> > The method that is proposed in Section 4.3 has also been explored for the general POMDP settings. For example, in [4,5,6], people have explored auxiliary losses (to predict the next observation, state, reward) for improving the representation learning of RNNs when learning optimal policies in POMDPs.
> >
>
> Thanks for pointing these related work out.  Here, we do not intend to claim this is an innovation, since, as you said, this is has been explored for the general POMDP settings. We just want to do some investigations here to help future studies, as part of our paper.
>
> We list it as a special Section because we think auxiliary predictive loss is an intuitive and important part of DOMDPs. It is intuitive solution to predict the delayed observation to solve DOMDPs. And there naturally raised two quesions:
>
> 1. **Choice of Representation:** Is it more effective to use encoded features directly or to rely on a predictive head?
> 2. **Network Architecture and Supervision:** Is it preferable to have a distinct prediction network, or should we RL loss for more effective supervision?
>
> We try to figure out when and where we can do it in Tab.1 and Tab.2 in Appendix G.1.1, which holds importance as reference for future studies.

---

> ### Author Response · Authors · 2023-11-23
> **Part 3**
>
> > “In the beginning of Section 3, baseline DRL methods are tested in delay signal settings and they are demonstrated to struggle in such a setting. Reasons are guessed for why they fail. However, I do not think that the failure of these baselines methods are studied well enough. I would like to see more ablation study on understanding how exactly they fail, which I believe will greatly help readers understand in what way DOMDPs are different to POMDPs and what are their unique challenges. Is it really due to the signal delay? Or like in general POMDPs, the RNNs cannot come up with good representations for decision making? For example, if these methods struggle to learn representations that support effective decision making, then one can test the accuracy of using these learned representations to predict the true current state. There are possible causes mentioned in paper but I find them rather vague and are not enough to motivate the proposed algorithmic changes.“
> >
>
> We appreciate the reviewers' suggestion to conduct a more thorough investigation into the failure of baseline methods in DOMDP settings. In response, we have performed additional experiments to better understand these failures and their implications for distinguishing DOMDPs from POMDPs. Our findings are summarized below:
>
> 1. **RNN can learn good representations.** We show additional experiment results to investigate in Sec. G.2 of the Appendix.  The representation loss in VRM with delay (e.g. even -1.505 with a 12-step delay) close to the -1.91 (when there is no delay), suggesting that representation learning is not the primary bottleneck.
>
>
>     | Delay | Walker2d | HalfCheetah | Hopper | Ant | Average |
>     | --- | --- | --- | --- | --- | --- |
>     | 0 | -1.87 ± 0.31 | -1.60 ± 0.15 | -2.50 ± 0.07 | -1.68 ± 0.21 | -1.91 |
>     | 1 | -1.90 ± 0.20 | -1.74 ± 0.10 | -2.54 ± 0.05 | -1.99 ± 0.43 | -2.04 |
>     | 2 | -2.25 ± 0.24 | -1.70 ± 0.20 | -2.62 ± 0.10 | -1.89 ± 0.21 | -2.10 |
>     | 4 | -2.26 ± 0.06 | -1.65 ± 0.22 | -2.49 ± 0.09 | -1.81 ± 0.24 | -2.05 |
>     | 8 | -1.92 ± 0.18 | -1.57 ± 0.08 | -1.81 ± 0.08 | -1.67 ± 0.20 | -1.74 |
>     | 12 | -1.86 ± 0.14 | -1.29 ± 0.20 | -1.56 ± 0.14 | -1.31 ± 0.10 | -1.50 |
> 2. **Redundant or insufficient information in input to the policy or critic network.**
>
>     Our new experimental results shows that the proper amount of historical information is important. Excessive history can negatively impact performance, as shown in the following table:
>
>     | Column is # of delay $\Delta t$; Row is Historical num $H$  $\downarrow$  | $\Delta t=0$ | $\Delta t=1$ | $\Delta t=4$ |
>     | --- | --- | --- | --- |
>     | $H=0$ | **97.7±24.0** | **87.8±22.3** | 47.6±20.3 |
>     | $H=1$ | **104.5±18.4** | **100.2±16.5** | **77.7±15.4** |
>     | $H=2$ | **98.8±23.4** | **93.7±18.9** | **84.3±16.3** |
>     | $H=4$ | **86.8±18.6** | 81.6±23.4 | **80.3±17.8** |
>     | $H=8$ | 68.3±12.5 | 73.0±13.6 | 58.6±15.5 |
>     | $H=12$ | 62.6±14.3 | 60.9±12.8 | 55.0±8.3 |
>
>     ** We **highlight** the lines in the error range of highest one, to be consistent with Tab.1.*
>
>     We investigate this by changing, $H$, the number of past actions of input to see how this affects the performance when delay $\Delta t$ is fixed to $\{0,1,4\}$.
>
>     Here, we present the result using both “Delay reconciled training for critic” and “State Augmentation - MLP”, but with different number of histories $H$. Note that he model is powered with all the effective tricks in our paper, with only the $H$ varies.
>
>     We can observe that incorporating no past actions ($H=0$) leads to suboptimal performance. For instance, with a delay of $\Delta t=4$, the absence of historical actions ($H=0$) resulted in a relatively low performance score of 47.6. Introducing a moderate amount of past actions significantly enhances performance, as evidenced by the improved score of 84.3. However, it's important to note that overloading the model with an excessive history (too high $H\geq 8$) reversely impacts the efficiency. This was clear when an even larger history led to a reduced performance of 55.0 ($H=12)$. Same trend can be found in all columns that the middle performance is higher than these of two sides.
>
>     This finding also provides a potential explanation for the limited success of RNN structures in our experiments, as RNNs inherently lack control over the extent of historical information utilized. So it brought redundant information such as state history and extra action history outside of $\Delta t$. The result also embraces our theoretical finding that only historical actions in $\Delta t$ is enough to keep the Markovian property in Theorem 4.1.
>
>     We hope this would also support our response to your another question, “common practice of POMDP that introducing historical state and actions” can not solve DOMDPs well.
>
>     This is an important observation, wee add new experimental results and analysis in Sec. G.3 of the Appendix.

---

> ### Author Response · Authors · 2023-11-23
> **Part 4**
>
> > “I also find lots of experimental details are missing from the paper, such as the architectures of networks, and the hyperparameters. Regarding hyperparameters, it would also be useful to know how are the hyperparameters tuned for each baseline and the proposed method to ensure fair comparison.”
> >
>
> Thank you for your feedback. We've addressed these concerns in Appendix Sec. [TODO] and summarize key points here:
>
> - **Network Architectures:** For all proposed algorithm variants, we maintained a consistent structure. We compared different architectures like RNNs, MLPs, and Transformers, ensuring similar parameter counts. For baselines, we strictly followed the architectures from their original source code.
> - **Hyperparameters:** We conducted extensive sweeps to ensure fair comparison. For each (algorithm, delay) pair, we select the best-performing hyperparameters.
>
> > “Also, for the baselines, such as DDPG, TD3, SAC, are they also RNN based? It would be strange to use a non-recurrent baseline in a POMDP environment. And are the past actions passed to the actors in these baseline methods? “
> >
>
> In Table 1, the versions of Vanilla DDPG, TD3, and SAC we used are not RNN-based. These vanilla implementations serve primarily to show the performance decline in DOMDP environment if nothing is done for the mainstream RL algorithms, rather than as competitive baselines.
>
> While we did provide the RNN based version, which is “RNN Strong Baseline [1]” (line 4th in Table.1). The author proposed to use RNN to beat other SOTA methods in POMDPs. So this implementation of RNN is very well tuned, thus capable to serve as a confident baseline of RNN version of SAC.
>
> [1] Recurrent model-free RL can be a strong baseline for many POMDP.
>
> > “In Figure 3, I find it weird to argue that baseline methods are failing by saying that there is a significant drop in performance from no delay to one-step delay. With different delay steps, we essentially have different POMDPs, which may inherently have different difficulty levels and different optimal performance. As such, the drop in performance I think is not sufficient to say the baseline methods are failing. However, a comparison to the optimal performance or performance from a better method would work.”
> >
>
> We acknowledge the points you've raised and have repainted the Fig.3:
>
> 1. **Consistent Delay Comparison:** We now display the performance of all methods under the same delay step in a single figure. This approach ensures a more equitable comparison by eliminating the variable of different delay steps, which, as you noted, can introduce varying levels of difficulty.
> 2. **New Performance Benchmark:** To provide a clearer evaluation of the baseline methods, we have added a benchmark line (dashed black) indicating the performance of our proposed method ("Ours") under the same delay. This serves as a reference for understanding the relative performance levels and the impact of the delay. We also mark the relative performance drop related to Ours.
>
> Please refer to the updated Figure 3 in the revised manuscript.
>
> We appreciate again for your comments, which help to improve the manuscript significantly.

---

### Official Review · Reviewer_dZmE · 2023-11-01

**Soundness:** 3 good
**Presentation:** 3 good
**Contribution:** 3 good
**Rating:** 6
**Confidence:** 5

**Summary:**

This paper considers the delay in MDP (e.g., inference delay, observation delay, action delay) and formulates Delayed-Observation MDP setting. Effective approach is proposed to deal with Delayed systems.

**Strengths:**

1. The studied problem is interesting and meaningful. Delayed systems in widely and well studied in control community, see [1] [2] for example. There are thousands of works addressing classic concerns such as system stability and safety. I would suggest the authors to survey and (briefly) compare more control papers in this field because control theory and reinforcement learning are highly related and share similar concerns.

2. The experimental results are encouraging and promising.

3. I think this paper is well-written and easy to follow and understand.


[1] Richard, J.P., 2003. Time-delay systems: an overview of some recent advances and open problems. automatica, 39(10), pp.1667-1694.
[2] Cacace, F., Germani, A. and Manes, C., 2010. An observer for a class of nonlinear systems with time varying observation delay. Systems & Control Letters, 59(5), pp.305-312.

**Weaknesses:**

My main concern is the real-world application. Delayed systems investigation is motivated from the real systems, so I think the approach/theory developed for delayed systems should be examined through real-world experiments such as robotic, autonomous vehicles rather than just artificial simulations.

**Questions:**

No question.

---

> ### Author Response · Authors · 2023-11-23
>
> > My main concern is the real-world application. Delayed systems investigation is motivated from the real systems, so I think the approach/theory developed for delayed systems should be examined through real-world experiments such as robotic, autonomous vehicles rather than just artificial simulations.
> >
>
> We value your emphasis on real-world applicability and acknowledge the importance of testing our approach in practical scenarios such as robotics and autonomous vehicles. However, due to constraints in time and available equipment, conducting experiments with actual robots and vehicles was not feasible within the rebuttal stage timeframe.
>
> To address this limitation and align our research closer to real-world condition, recognizing a key difference between MoJoCo's deterministic nature and the inherent randomness of real-world settings, we have introduced elements of randomness into our simulations. This adjustment aims to bridge the gap between our controlled environment and the unpredictability of real-world applications.”
>
> | Noise Scale | Method | 0 | 1 | 2 | 4 | 8 | 12 |
> | --- | --- | --- | --- | --- | --- | --- | --- |
> | $\sigma=0.05$ | Ours$^*$ | 32.8±7.4 (100%) | 35.0±7.9 (106.71%) | 32.9±5.7 (100.30%) | 31.7±5.8 (96.65%) | 27.1±7.2 (82.62%) | 26.1±6.0 (79.57%) |
> | $\sigma=0.05$ | Vanilla SAC | 32.8±7.4 (100%) | 12.4±3.9 (37.80%) | 11.0±1.9 (33.54%) | 9.7±1.5 (29.57%) | 8.1±1.1 (24.70%) | 7.8±1.3 (23.78%) |
> | $\sigma=0.1$ | Ours$^*$ | 25.5±2.9 (100%) | 23.1±4.0 (90.59%) | 24.3±5.3 (95.29%) | 21.2±4.7 (83.14%) | 21.9±6.8 (85.88%) | 21.2±4.2 (83.14%) |
> | $\sigma=0.1$ | Vanilla SAC | 25.5±2.9 (100%) | 11.0±1.4 (43.14%) | 10.4±1.7 (40.78%) | 9.4±1.3 (36.86%) | 7.4±1.2 (29.02%) | 6.8±3.3 (26.67%) |
> | $\sigma=0.2$ | Ours$^*$ | 14.2±1.2 (100%) | 13.3±1.9 (93.66%) | 13.3±1.8 (93.66%) | 13.4±2.1 (94.37%) | 11.8±2.3 (83.10%) | 13.1±2.5 (92.25%) |
> | $\sigma=0.2$ | Vanilla SAC | 14.2±1.2 (100%) | 10.9±1.9 (76.76%) | 10.1±1.1 (71.13%) | 8.6±1.5 (60.56%) | 8.0±1.2 (56.34%) | 4.6±5.0 (32.39%) |
> | $\sigma=0.4$ | Ours$^*$ | 11.3±1.7 (100%) | 11.0±1.4 (97.35%) | 10.0±2.3 (88.50%) | 10.8±2.1 (95.58%) | 10.7±1.7 (94.69%) | 11.1±1.1 (98.23%) |
> | $\sigma=0.4$ | Vanilla SAC | 11.3±1.7 (100%) | 10.0±1.2 (88.50%) | 10.1±1.2 (89.38%) | 8.8±2.1 (77.88%) | 8.0±1.8 (70.80%) | 5.0±5.4 (44.25%) |
>
> $^*$ We choose “State Augmentation - MLP” in Tab.1 as an example implementation of “Ours”.
>
> ** In our results table, each cell contains **`{Value}({Relative%})`**, where **`{Value}`** is normalized in the same manner as in Tab.1 of our paper. The **`{Relative%}`** indicates the relative reward compared to the highest value of each line.
>
> We can see that the performance of Vanilla SAC significantly decreases with the introduction of delay, while our method substantially improves performance in delayed scenarios.
>
> We understand that this is not a real-world environment, while we did make progress to make our experiment can be close to real-world.
>
> 1. We study both fixed and unfixed random delay (Tab.1, Tab.2)
> 2. We study probabilistic environment. e.g. noisy and sticky actions (G.1.3), and noisy version of Mujoco above (have been add to G.1.2).
>
>
> We would like to emphasize that all results presented in our paper, as well as those highlighted in the response above, **consistently** demonstrate that our performance is **significantly** superior to other baselines.
>
> We hope these updates and clarifications will address your concerns to some extent. We remain committed to improve our work by considering your valuable suggestions.
>
> > “ I would suggest the authors to survey and (briefly) compare more control papers in this field because control theory and reinforcement learning are highly related and share similar concerns.”
> >
>
> We agree that the control related work worth investigation in the DOMDP topic, we will take time to do a comprehensive survey and add discussion in the paper.
>
> Thanks for your comments!

---

### Official Review · Reviewer_aPPk · 2023-11-05

**Soundness:** 3 good
**Presentation:** 3 good
**Contribution:** 3 good
**Rating:** 6
**Confidence:** 3

**Summary:**

This study tackles a significant yet frequently underestimated issue in the integration of reinforcement learning agents into real-world applications: signal delay. The authors establish a formal framework for this problem by framing it as delayed-observation Markov decision processes. They then introduce a novel algorithm designed to effectively address this challenge. The empirical findings from experiments reveal that the proposed method consistently demonstrates strong performance, even in scenarios with multiple steps of signal delay. This research sheds light on a crucial aspect of reinforcement learning in practical settings, offering a promising solution for mitigating the impact of signal delays on agent performance.

**Strengths:**

1. Clarity and Readability: The manuscript is well written, making it easy for readers to comprehend the presented concepts and findings. The clear and coherent writing style enhances the accessibility of the research.

2. Significant Problem Addressed: This paper takes on an important and frequently underestimated issue, which has often been overlooked in prior research.

3. Impressive Performance: The achieved performance, when compared to conventional vanilla algorithms, is notably robust and impressive. This showcases the effectiveness and practicality of the proposed approach, making it a noteworthy contribution to the field.

**Weaknesses:**

One potential weakness of the paper lies in its handling of experiments in a probabilistic state transition setting. While the study explores action space perturbation, such as sticky action and noisy action, it could benefit from a more comprehensive consideration of the stochasticity within the state space. This might involve scenarios with dynamic background images or robots with joint angular positions sampled from a normal distribution with high variance. Incorporating these additional sources of uncertainty would provide a more realistic and robust evaluation of the proposed algorithm's performance under varied conditions.

**Questions:**

1. Could the performance of the method be further enhanced by considering advanced world models, like the one developed for model-based RL in Dreamer[1], as an alternative for the architecture of the state augmentation and prediction module?

[1] Hafner, Danijar, et al. "Mastering diverse domains through world models." arXiv preprint arXiv:2301.04104 (2023).

---

> ### Author Response · Authors · 2023-11-23
>
> > One potential weakness of the paper lies in its handling of experiments in a probabilistic state transition setting. While the study explores action space perturbation, such as sticky action and noisy action, it could benefit from a more comprehensive consideration of the stochasticity within the state space. This might involve scenarios with dynamic background images or robots with joint angular positions sampled from a normal distribution with high variance. Incorporating these additional sources of uncertainty would provide a more realistic and robust evaluation of the proposed algorithm's performance under varied conditions.
> >
>
> Thank you for your insightful suggestion regarding the importance of investigating probabilistic state transitions to enhance the comprehensiveness and realism of our study.
>
> In response, we have modified the Mujoco environment to introduce diagonal Gaussian noise, denoted as $\mathcal{N}(0,\sigma^2)$, to the state at each step.
>
> | Noise Scale | Method | 0 | 1 | 2 | 4 | 8 | 12 |
> | --- | --- | --- | --- | --- | --- | --- | --- |
> | $\sigma=0.05$ | Ours$^*$ | 32.8±7.4 (100%) | 35.0±7.9 (106.71%) | 32.9±5.7 (100.30%) | 31.7±5.8 (96.65%) | 27.1±7.2 (82.62%) | 26.1±6.0 (79.57%) |
> | $\sigma=0.05$ | Vanilla SAC | 32.8±7.4 (100%) | 12.4±3.9 (37.80%) | 11.0±1.9 (33.54%) | 9.7±1.5 (29.57%) | 8.1±1.1 (24.70%) | 7.8±1.3 (23.78%) |
> | $\sigma=0.1$ | Ours$^*$ | 25.5±2.9 (100%) | 23.1±4.0 (90.59%) | 24.3±5.3 (95.29%) | 21.2±4.7 (83.14%) | 21.9±6.8 (85.88%) | 21.2±4.2 (83.14%) |
> | $\sigma=0.1$ | Vanilla SAC | 25.5±2.9 (100%) | 11.0±1.4 (43.14%) | 10.4±1.7 (40.78%) | 9.4±1.3 (36.86%) | 7.4±1.2 (29.02%) | 6.8±3.3 (26.67%) |
> | $\sigma=0.2$ | Ours$^*$ | 14.2±1.2 (100%) | 13.3±1.9 (93.66%) | 13.3±1.8 (93.66%) | 13.4±2.1 (94.37%) | 11.8±2.3 (83.10%) | 13.1±2.5 (92.25%) |
> | $\sigma=0.2$ | Vanilla SAC | 14.2±1.2 (100%) | 10.9±1.9 (76.76%) | 10.1±1.1 (71.13%) | 8.6±1.5 (60.56%) | 8.0±1.2 (56.34%) | 4.6±5.0 (32.39%) |
> | $\sigma=0.4$ | Ours$^*$ | 11.3±1.7 (100%) | 11.0±1.4 (97.35%) | 10.0±2.3 (88.50%) | 10.8±2.1 (95.58%) | 10.7±1.7 (94.69%) | 11.1±1.1 (98.23%) |
> | $\sigma=0.4$ | Vanilla SAC | 11.3±1.7 (100%) | 10.0±1.2 (88.50%) | 10.1±1.2 (89.38%) | 8.8±2.1 (77.88%) | 8.0±1.8 (70.80%) | 5.0±5.4 (44.25%) |
>
> *“Ours” is “State Augmentation - MLP” in Tab.1
>
> ** {Value}({Relative%}), where {Value} is normalized in the same manner as in Tab.1 of our paper. The {Relative%} indicates the relative reward compared to the highest value of each line.
>
> The results comparing Ours$^*$ with the Vanilla SAC are presented below. It is important to note that when delay=0, Vanilla SAC is equivalent to Ours$^*$. Key observations from our experiments are as follows:
>
> 1. The performance of Vanilla SAC significantly decreases with the introduction of delay, dropping from 100% to 23.78%, 26.67%, 32.39% and 44.25% when the  $\sigma$ is 0.05, 0.1, 0.2, 0.4, respectively.
> 2. Our method substantially improves performance in delayed scenarios. e.g. when $\Delta t=12$, the performance increase is 23.78%→79.57%, 26.67%→83.14%, 32.39%→92.25%, 44.25%→98.23%, when the $\sigma$ is 0.05, 0.1, 0.2, 0.4, respectively.
>
> These observations are consistent with those in the original Mujoco environment, demonstrating that our improvements are both consistent and significant.
>
> While we acknowledge that this may not encompass all probabilistic environments, we wish to highlight:
>
> 1. The **significant** and **consistent** improvement demonstrated in the above table.
> 2. The inclusion of two additional probabilistic environments discussed in Section G.1.2 of our paper.
>
> These consistent results strongly support the efficacy of our method in probabilistic environments.
>
> Furthermore, it is important to note that our theoretical framework consider general probabilistic scenario as in Theorem 2.1 and Theorem 4.1.
>
> We hope these experimental and theoretical results adequately address your concerns.
>
> We appreciate your suggestion and have included the Gaussian probabilistic experiment in Section G.1 of the paper.
>
> > Could the performance of the method be further enhanced by considering advanced world models, like the one developed for model-based RL in Dreamer[1], as an alternative for the architecture of the state augmentation and prediction module?
> >
>
> Good point! Actually, we did preliminary experiments about using advanced world models in this research, in particular the recurrent state space model (RSSM) used in Dreamer [1]. The problem of using RSSM is that since it models single-step state-transition, the prediction error may accumulate (especially due to a stochastic nature of RSSM) in the scenario of multi-steps delay. The empirical results also failed to support the effectiveness of using RSSM. Rather, MLP network is simple and good enough. Nonetheless, we appreciate this comment and will consider more powerful world models in future exploration.
>
> [1] Mastering diverse domains through world models.

---

### Official Review · Reviewer_eRXT · 2023-11-07

**Soundness:** 4 excellent
**Presentation:** 3 good
**Contribution:** 4 excellent
**Rating:** 8
**Confidence:** 3

**Summary:**

The authors study the impact of signal delay in reinforcement learning problems. Their results show that even minor delays in receiving states can negatively impact the performance of various existing deep-learning algorithms. The authors formalize delay as the Delayed Observation Markov Decision Process and show theoretically grounded means to modify existing actor-critic algorithms to account for the delay. They propose augmenting the state with previous actions and consider conditioning on reconstructed observations at deployment. Their experiments suggest combining both techniques can mitigate the effects of signal delay.

**Strengths:**

Overall, the paper is well written. The authors succinctly characterize the issues of observation delay and motivate it thoroughly across various domains. Each section transitions well to the next and is accessible to readers. We appreciate the author's approach of formalizing the problem and then providing general solutions. Identifying a missing property in RL algorithms and generally applicable techniques is a good approach, and the performed experiments offer compelling evidence of their efficacy. We feel the authors do due diligence with their investigations to validate their solutions, and we appreciate them focusing on accessible solutions and ablating them in the paper.

**Weaknesses:**

One issue with the paper is the proposed use of previous actions in the augmented state representation. Some previous work suggests, particularly for sequence models [1], that this can be problematic, and it seems the results may mean this is the case as the Transformer and RNN results seem to benefit less from the state augmentation approach, especially as the horizons increase. We consider this a future work that the authors might consider discussing. Theorem 4.1 might require a few additional words to emphasize whether the provided proof is from previous work, mainly as the authors cite a paper when talking about this theorem.

Although we overall find the paper accessible, some parts could be better written or expand on important details. Specifics below:

1.)  Section 3 could benefit from clarifying the experimental setup or informing the reader where to find such information, for example. Additional comments below:

2.) Including the % error cap for Figure 3 d. Consider including information on the calculations for these are done.

3.) Figure 4 needs to be clarified what the robot is observing. Perhaps some thought bubble or other intermediate object to distinguish where the robot is versus where it THINKS it might help if, for example, if s_t - \Delta T is supposed to be the information the robot is currently making decisions from, that was unclear.

4.) Consider moving Table 1 to the appendix and use plots to highlight trends the authors observed from the results. It's difficult to process and follow all the observations.

5.) Figure 6 is a bit difficult to dissect the details of the work. I suggest adding additional annotations to plots to make it more transparent what each subplot visualizes.


Citation:
[1] Shaking the foundations: delusions in sequence models for interaction and control. Ortega et al. 2021.

**Questions:**

1.) Aren't delayed rewards already accounted for via the value function? Consider removing this wording in the introduction or clarify that your research looks at delays in states, as we could not find a discussion on addressing delays in received rewards.
2.) The transition probability in section 2.2 is confusing. The next state only transitions if s’^{-t+ = s^{-t+1} and otherwise would return 0 if any states don't match. Is this correct, or a misunderstanding?
3.) What were the parameters used in the experiments of Section 3? Including experimental details is reasonable for reproducibility. If they are in the appendix, please mention this in the section
4.) What about including the time since the last received sample as a state feature?

---

> ### Author Response · Authors · 2023-11-23
> **Part 1**
>
> > One issue with the paper is the proposed use of previous actions in the augmented state representation. Some previous work suggests, particularly for sequence models [1], that this can be problematic, and it seems the results may mean this is the case as the Transformer and RNN results seem to benefit less from the state augmentation approach, especially as the horizons increase. We consider this a future work that the authors might consider discussing. Theorem 4.1 might require a few additional words to emphasize whether the provided proof is from previous work, mainly as the authors cite a paper when talking about this theorem.
> >
>
> Thank you for your valuable feedback. We appreciate the opportunity to look into the concerns you've raised.
>
> You mentioned the issue of using past actions in our model. We understand that this can be a double-edged sword:
>
> 1. **Good Point**: Including past actions helps make our model's environment more predictable and easier to understand.
> 2. **Downside**: As pointed out in [1], this approach can sometimes cause problems.
>
> To explore this further, we experimented with changing, $H$, the number of past actions our model considers. We wanted to see how this affects the performance when delay $\Delta t$ is fixed to $\{0,4\}$ . Here's a summary of what we found:
>
> | Column is # of delay $\Delta t$; Row is Historical num $H$  $\downarrow$  | $\Delta t=0$ | $\Delta t=1$ | $\Delta t=4$ |
> | --- | --- | --- | --- |
> | $H=0$ | **97.7±24.0** | **87.8±22.3** | 47.6±20.3 |
> | $H=1$ | **104.5±18.4** | **100.2±16.5** | **77.7±15.4** |
> | $H=2$ | **98.8±23.4** | **93.7±18.9** | **84.3±16.3** |
> | $H=4$ | **86.8±18.6** | 81.6±23.4 | **80.3±17.8** |
> | $H=8$ | 68.3±12.5 | 73.0±13.6 | 58.6±15.5 |
> | $H=12$ | 62.6±14.3 | 60.9±12.8 | 55.0±8.3 |
>
> ps. We **highlight** the lines in the error range of highest one, to be consistent with Tab.1.
>
> ps. We use State Augmentation - MLP in Tab.1 as method for above table.
>
> - When we don't consider any past actions ($H=0$), and there's no delay in the data ($Δt=0$), the model performs quite well. But if we start adding too many past actions (like $H=4,8,12$), the performance drops significantly, result in 62.6% performance when $H=12$.
> - In scenarios where there's a delay in the data ($Δt>0$), we observed that not incorporating any past actions ($H=0$) leads to suboptimal performance. For instance, with a delay of $\Delta t=4$, the absence of historical actions ($H=0$) resulted in a relatively low performance score of 47.6. Introducing a moderate amount of past actions significantly enhances performance, as evidenced by the improved score of 84.3. However, it's important to note that overloading the model with an excessive history ($H$ too high) reversely impacts the efficiency. This was clear when an even larger history led to a reduced performance of 55.0. This pattern underscores the necessity of a balanced approach in incorporating historical actions, especially in the context of delayed data.
>
> In summary, our findings indicate that optimal performance is generally achieved when the number of past actions considered, $H$, closely aligns with the delay in the data, $\Delta t$. RNN
>
> This finding also provides a potential explanation for the limited success of RNN structures in our experiments, as RNNs inherently lack control over the extent of historical information utilized, it just takes all historical states and actions into account. So it brought redundant information such as state history and extra action history outside of $\Delta t$, distract the model from values information.
>
> We have added the results and discussion to Sec.G.3. Thanks for your comments.
>
> [1] Shaking the foundations: delusions in sequence models for interaction and control. Ortega et al. 2021.

---

> ### Author Response · Authors · 2023-11-23
> **Part 2**
>
> > 1.) Aren't delayed rewards already accounted for via the value function? Consider removing this wording in the introduction or clarify that your research looks at delays in states, as we could not find a discussion on addressing delays in received rewards.
> >
>
> Thanks for your kind advise. We removed “reward” part in the introduction.
>
> > 2.) The transition probability in section 2.2 is confusing. The next state only transitions if $s’^{-t+ = s^{-t+1}}$ and otherwise would return 0 if any states don't match. Is this correct, or a misunderstanding?
> >
>
> Thanks for pointing this out this potential confusing point. The superscript $(-t)$ on the state $s$ is the relative timestep to the current one. And the transition probability function discribes “one step shift” of all the $s$ in $\sigma$ for each step transition. This is why $\mathbb{I}({s'^{(-t)} = s^{(-t+1)}}) = 1$ only if $s'$ is the next state of $s$ and $=0$ otherwise. This means exact one-step shift of for every $s^\{(-t)\}$ in $\sigma$.
>
> > 3.) What were the parameters used in the experiments of Section 3? Including experimental details is reasonable for reproducibility. If they are in the appendix, please mention this in the section
> >
>
> We add it to Sec.E in the appendix and reference them in the Sec.5.1, the start of the experiment results.
>
> > 4.) What about including the time since the last received sample as a state feature?
> >
>
> Thanks for your suggestion.
>
> We think that this will take effect under case that the delay is unfixed, append time with the sample is promise to improve the performance.
>
> 1. According to our reply to your first question, we know that the when historical number $H$ is equal to delay $\Delta t$, the performance is the best. So it is promising that timestamp can help to focus on the historical in $\Delta t$ range, even when the delay is unfixed.
> 2. Especially, when the state is from multiple source, e.g. robot with multiple sensors, financial agent with multiple network information source, losing the time stamping information can make it hard to make decisions.
>
> However, it's important to note that the effectiveness of this method may vary depending on the specific environment. For instance, certain sensors might not support time stamping, or network messages may lack time-related data. This make the trick is not a general solution for all cases
>
> We create a new section Sec.D in the appendix to discuss this in detail.

---

### Author Response · Authors · 2023-11-23
**General Response**

We greatly appreciate all reviewers for your time and effort in providing these insightful feedback that help us improve our work. We have submitted a revised version of the paper that highlights the changes in red color. We summarize the changes as follows (the section numbers correspond to the revised version):

1. `[eRXT]` We added an experiment investigating the failure of RNNs to Appendix G.3. The discussion on "adding the time since the last received sample" has been incorporated into Appendix D. Additionally, implementation details have been included in Appendix E.
2. `[aPPk][dZmE]` In Appendix G.1, we have introduced an experiment on Mujoco, with state sampled from a normal distribution, to offer futher performance demonstration of our techniques in a probabilistic environment `[aPPk]`. Also, this addition aims to provide a simulation closer to real-world scenarios `[dZmE]`.
3. `[6hPB]` We have revisited the investigation of RNN failures in Appendix G.3. The feature representation learned has been examined in Appendix G.2. We have expanded implementation details in Appendix E, specifically focusing on hyper parameter selection in E.3 and network architecture in E.2. Sections 4.1 and 4.3 now address related work more comprehensively. Finally, Figure 3 has been revised for enhanced clarity.

---

### Meta-Review · Area_Chair_7m6x · 2023-12-11

**Metareview:**

This paper formalizes a decision-making problem to match reality where the observations, impact of actions, or rewards are delayed before being given to the agent. The paper then shows that the techniques for solving partially observable environments are ineffective at dealing with delayed signals. A principled technique for dealing with signal delay is introduced, and its effectiveness is demonstrated. While the results are only for simulation, the paper accomplishes its goal and is of interest to the community. I recommend the paper for acceptance.

**Justification For Why Not Higher Score:**

I recommend this paper for a spotlight as some but not all of the community will be interested in the topic.

**Justification For Why Not Lower Score:**

I think this paper is worthy of a spotlight as it is well presented and will furthers the understanding of how one needs to model and account for different components of real-world environments.

---

### Decision · Program_Chairs · 2024-01-16

Accept (spotlight)